# Inefficient nitrogen transport to the lower mantle by sediment subduction

Weihua Huang [1], Yan Yang [1] ✉, Yuan Li [2,3], Zheng Xu [2], Shuiyuan Yang[4], Shengbin Guo[4] & Qunke Xia [1]

The fate of sedimentary nitrogen during subduction is essential for understanding the origin of nitrogen in the deep Earth. Here we study the behavior of nitrogen in slab sediments during the phengite to K-hollandite transition at 10–12 GPa and 800–1100 °C. Phengite stability is extended by 1–3 GPa in the nitrogen ($NH_4^+$)-bearing system. The phengite-fluid partition coefficient of nitrogen is 0.031 at 10 GPa, and K-hollandite-fluid partition coefficients of nitrogen range from 0.008 to 0.064, showing a positive dependence on pressure but a negative dependence on temperature. The nitrogen partitioning data suggest that K-hollandite can only preserve ~43% and ~26% of the nitrogen from phengite during the phengite to K-hollandite transition along the cold and warm slab geotherms, respectively. Combined with the slab sedimentary nitrogen influx, we find that a maximum of ~$1.5 \times 10^8$ kg/y of nitrogen, representing ~20% of the initial sedimentary nitrogen influx, could be transported by K-hollandite to the lower mantle. We conclude that slab sediments may have contributed less than 15% of the lower mantle nitrogen, most of which is probably of primordial origin.

As one of the most abundant elements in the solar system, nitrogen (N) is not only essential for life but also affects paleoclimate and habitability of the Earth[1–4]. Although nitrogen has a high volume proportion (78%) of the Earth's present atmosphere, the deep Earth has been suggested to be a major nitrogen reservoir[5], based on the nitrogen storage capacities of mantle minerals[6–11]. However, the origin and evolution of mantle nitrogen remains an ongoing matter of debate[3,5,12–17]. To address this question, the recycling efficiency of nitrogen to the deep Earth through plate subduction has been extensively explored, but still with inconsistent results[17]. Based on nitrogen contents and isotopes of subducted sediments and oceanic crust, some studies suggested a significant nitrogen loss to the sub-arc depth during plate subduction[18–22]. On the contrary, other studies proposed that considerable nitrogen could be delivered to the deep mantle beyond the sub-arc depth[23–30].

Sediments have the highest nitrogen contents in subduction slabs[31]. Nitrogen is mainly incorporated as ammonium ($NH_4^+$) in

K-bearing minerals[6,32], and thus the stabilities of ammonium-bearing silicates and nitrogen partitioning between minerals and fluids/melts dictate the amounts and depth of nitrogen delivery during subduction. To date, several experimental studies have explored the partitioning behavior of nitrogen in sediments during slab dehydration or melting. The nitrogen partitioning behavior of micas in the subducted sediment under reduced conditions indicates a high retention of nitrogen in cold slabs[33]. Based on the influences of geothermal gradient and oxygen fugacity on the nitrogen partitioning between biotite/K-feldspar and melt/fluid in the sediment system, it was further highlighted that warm and oxidizing slabs were inefficient for nitrogen transport to the deep mantle[34]. Besides, a higher nitrogen content would decrease the nitrogen partition coefficients between silicate phases (biotite/melt) and aqueous fluid[35]. The nitrogen partitioning between phengite and aqueous fluid/supercritical fluid of two recent studies reveals an important role of phengite in nitrogen transport to the deep Earth in

[1]Key Laboratory of Geoscience Big Data and Deep Resource of Zhejiang Province, School of Earth Sciences, Zhejiang University, Hangzhou, China. [2]State Key Laboratory of Isotope Geochemistry, Guangzhou Institute of Geochemistry, Chinese Academy of Sciences, Guangzhou, China. [3]Bayerisches Geoinstitut, Universität Bayreuth, Bayreuth, Germany. [4]State Key Laboratory of Geological Processes and Mineral Resources, China University of Geosciences, Wuhan, China. ✉e-mail: yanyang2005@zju.edu.cn

hot slabs[36,37]. Overall, these previous studies have provided important constraints for understanding the nitrogen transport efficiency by mica in sediments, but with the experimental conditions mainly limited to the sub-arc depth. Whether and how much nitrogen could be transported to the deep mantle beyond the sub-arc depth by sediment subduction is poorly known.

K-hollandite, the product of phengite breakdown, could be stable to the lower mantle[38] and has been found to have a large nitrogen storage capacity[6]. In this study, we experimentally study the behavior of nitrogen after the transition of phengite to K-hollandite at 10–12 GPa and estimate the deep subduction efficiency of sedimentary nitrogen to the depth below 300 km. Our new mineral-fluid partitioning data suggest that the phengite to K-hollandite transition could play as a significant nitrogen filter, allowing for sediments carrying a limited amount of nitrogen into the lower mantle. Our results provide new insights into the contribution of sediment subduction to the origin of nitrogen in the lower mantle.

## Results and discussion

### Mineral assemblages

Nine experiments were conducted on volatile-rich pelite, including eight experiments (NH$_3$-bearing runs) starting with NH$_3$ (25 wt%)·H$_2$O solution at 10–12 GPa and 800–1100 °C for 48–72 h, and one experiment (NH$_4$NO$_3$-bearing run) using NH$_4$NO$_3$ as the starting volatile at 11 GPa and 1000 °C for 72 h (Table 1). In each experiment, the silicate and volatile mass ratio is 12–15 (Supplementary Table 1). The produced mineral assemblage at 10 GPa and 1000 °C included phengite, garnet, clinopyroxene, stishovite, and kyanite (Fig. 1a). At 10–11 GPa and 1100 °C, phengite disappeared, but K-hollandite occurred (Fig. 1b). Under pressures of 10.5–12 GPa and temperatures of 800–1000 °C, the phase assemblages transformed into K-hollandite + garnet + clinopyroxene + stishovite + topaz-OH ± Ti-oxides (Fig. 1c). For the NH$_4$NO$_3$-bearing run at 11 GPa and 1000 °C, the mineral assemblage was K-hollandite + garnet + clinopyroxene + stishovite + kyanite + hematite + Fe-Ti phase (Fig. 1d). The detailed chemical compositions of the run products are provided in Supplementary Text 1, Supplementary Fig. 1 and Supplementary Data 1.

### Stability of phengite in the presence of reduced nitrogen

According to the mineral assemblages of the NH$_3$-bearing runs, the boundary of phengite to K-hollandite transition has a negative slope (Fig. 2a). The transition in the nitrogen-bearing system is compared with that in the nitrogen-free system of the same chemical compositions reported by Ono[39]. (Fig. 2a), which shows that the stability field of phengite in the NH$_3$-bearing system is extended by 1–3 GPa. Figure 2b shows that the substitution of (Mg, Fe$^{2+}$)$^{VI}$ + Si$^{IV}$ for Al$^{VI}$ + Al$^{IV}$ in phengite

displays a strong correlation with pressure in the pelitic system, and the Si, Al, Mg, and Fe contents of phengite in this study and the study of Ono[39]. have the same pressure dependence. This implies that the observed extension of the stability field of phengite in the NH$_3$-bearing system should not have been caused by the uncertainty of our experimental pressure derived from pressure calibration of the multi-anvil apparatus (Methods). The possible impact of Si/Al ratio on the stability of phengite[40] can also be ruled out here because the contents of Si and Al in phengite and K-hollandite have no difference between the systems with and without nitrogen (Fig. 2b, c). The extended stability field of phengite in the NH$_3$-bearing system should be caused by the presence of NH$_4^+$ in phengite. It has been reported that nitrogen, located as ammonium in the interlayer site of phengite (Supplementary Fig. 2), can form hydrogen bonding with the basal oxygen of the tetrahedral[41]. Huang et al.[42] showed that the formation of such hydrogen bonding can prohibit lattice weakening of phengite (with ~2000 ppm NH$_4^+$) and thus favor the stabilization of phengite at high temperatures. Huang et al.[43] also showed that even trace amounts of NH$_4^+$ (<1000 ppm[44]) can enhance the thermal stability of muscovite. There is also other evidence to show that the presence of volatiles with low concentrations can affect the mineral phase transitions. For example, Grützner et al.[45] showed that the presence of ~1500–4000 ppm F in olivine and wadsleyite can increase the pressure for the olivine to wadsleyite transition by ~2.6 GPa compared to the volatile-free system. Therefore, it is reasonable to conclude that the extended stability field of phengite in the NH$_3$-bearing system results from the presence of NH$_4^+$ in phengite, and this conclusion means that in slab sediments the NH$_4^+$-bearing phengite is more stable, probably up to 10 GPa, than the NH$_4^+$-free phengite.

### Nitrogen contents and partitioning between phengite, K-hollandite, and fluid

Based on the measured nitrogen contents of the produced minerals (Supplementary Data 1), phengite and K-hollandite are the major nitrogen hosts in the NH$_3$-bearing runs. At 10 GPa and 1000 °C, phengite contains ~4891 ± 975 ppm nitrogen. In the rest of the NH$_3$-bearing runs that produced K-hollandite, K-hollandite contains 1020 ± 399 to 8919 ± 1164 ppm nitrogen. Figure 3a, b shows that the nitrogen content of K-hollandite decreases with increasing temperature but increases with increasing pressure, which can be described as the following equation:

$$\log\left(C_N^{K-holl}, ppm\right) = \frac{-2481.25(\pm 1539.39)}{T} + \frac{482.05(\pm 127.99)P}{T} + 1.42(\pm 0.57)(R^2 = 0.82, p-value = 0.01) \quad (1)$$

**Table 1 | Run conditions, products, and the calculated nitrogen partition coefficients between phengite, K-hollandite, and fluid**

| Runs[a] | P (GPa) | T (°C) | t (h) | Phases[b] | Solutes(wt%)[c] | N in Phe and K-holl (wt%) | N in fluid (wt%)[d] | D$_N^{Phe/Fluid}$ | D$_N^{K-holl/Fluid}$ |
|---|---|---|---|---|---|---|---|---|---|
| NH-1 | 10 | 1000 | 48 | Phe, Cpx, Grt, Ky, St | 25–52 | 0.49 (Phe) | 13.4–19.4 | 0.031 (8) | – |
| NH-2 | 10 | 1100 | 48 | K-holl, Cpx, Grt, Ky, St | 28–55 | 0.10 (K-holl) | 10.8–16.3 | – | 0.008 (2) |
| NH-3 | 10.5 | 800 | 48 | K-holl, Cpx, Grt, Tp-OH, St | 19–42 | 0.70 (K-holl) | 13.7–18.0 | – | 0.045 (9) |
| NH-4 | 11 | 800 | 48 | K-holl, Cpx, Grt, Tp-OH, St | 19–41 | 0.83 (K-holl) | 13.5–17.6 | – | 0.054 (10) |
| NH-5 | 11 | 900 | 48 | K-holl, Cpx, Grt, Tp-OH, St | 20–42 | 0.89 (K-holl) | 13.2–17.4 | – | 0.059 (12) |
| NH-6 | 11 | 1000 | 72 | K-holl, Cpx, Grt, Tp-OH, St | 25–51 | 0.66 (K-holl) | 12.4–17.8 | – | 0.045 (11) |
| NH-7 | 11 | 1100 | 48 | K-holl, Cpx, Grt, Ky, St | 28–55 | 0.29 (K-holl) | 10.5–15.7 | – | 0.023 (6) |
| NH-8 | 12 | 1000 | 48 | K-holl, Cpx, Grt, Tp-OH, St | 26–51 | 0.84 (K-holl) | 11.0–15.8 | – | 0.064 (16) |
| N2-1 | 11 | 1000 | 72 | K-holl, Cpx, Grt, Ky, St, Hm, Fe-Ti phase | 25–51 | b.d. (K-holl) | – | – | – |

[a]NH runs use NH$_3$ solution as nitrogen source, while N$_2$ run uses NH$_4$NO$_3$ as nitrogen source.
[b]Phases with abundances above 1 wt%.
[c]Solutes in supercritical fluids are predicted by machine learning model (Extra Trees).
[d]Fluid consists of water, nitrogen, and solutes.

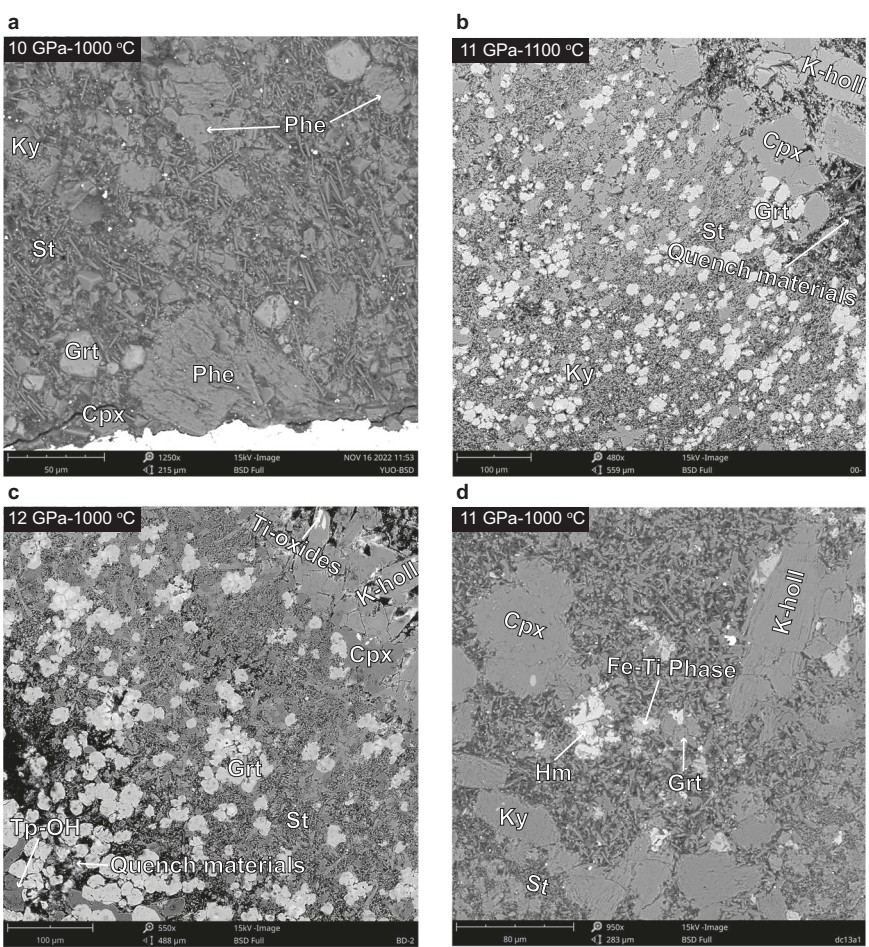

**Fig. 1 | Representative back-scattered electron (BSE) images of the run products. a–c** $NH_3$-bearing runs. **d** $NH_4NO_3$-bearing run. phengite (Phe), K-hollandite (K-holl), clinopyroxene (Cpx), garnet (Grt), kyanite (Ky), topaz-OH (Tp-OH), stishovite (St), hematite (Hm).

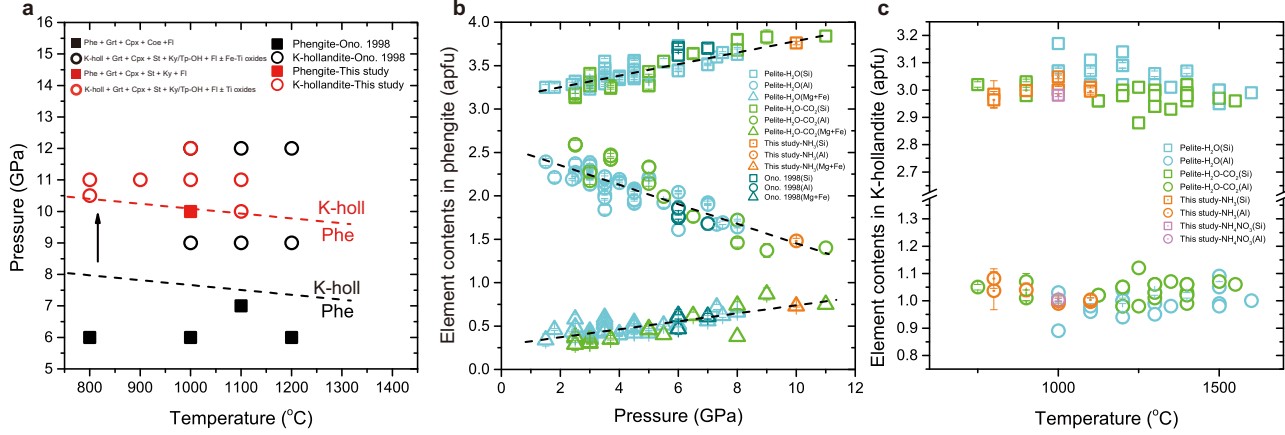

**Fig. 2 | Comparison of the stabilities and chemical compositions of the run products between our study and previous studies. a** Phase diagrams for nitrogen-bearing and nitrogen-free slab sediments at 6–12 GPa. **b** Chemical compositions of phengite vs. pressures and (**c**) K-hollandite vs. temperatures. Error bars represent the standard deviation based on Monte Carlo simulations that compound uncertainties for oxides (wt%). The source data for the $NH_3$-bearing (orange) and $NH_4NO_3$-bearing (magenta) systems are provided in Supplementary Data 1. The data for the pelite-$H_2O$ system (cyan) are from refs. [39,79–81] and the data for the pelite-$H_2O$-$CO_2$ system (green) are from refs. [81–84].

where $C_N^{K-holl}$(ppm) is the nitrogen content of K-hollandite in ppm, T is temperature in K, and P is pressure in GPa. However, in the $NH_4NO_3$-bearing run, no nitrogen was detected in the K-hollandite (Supplementary Text 1), in contrast to the high nitrogen content of 6553 ± 813 ppm in the K-hollandite of the $NH_3$-bearing run at the same P-T conditions. This suggests that high oxygen fugacity in the $NH_4NO_3$-bearing run, as indicated by the presence of hematite, stabilizes $N_2$ over reduced nitrogen species, and nitrogen is stored in K-hollandite mainly as ammonium (Supplementary Text 1 and see more below).

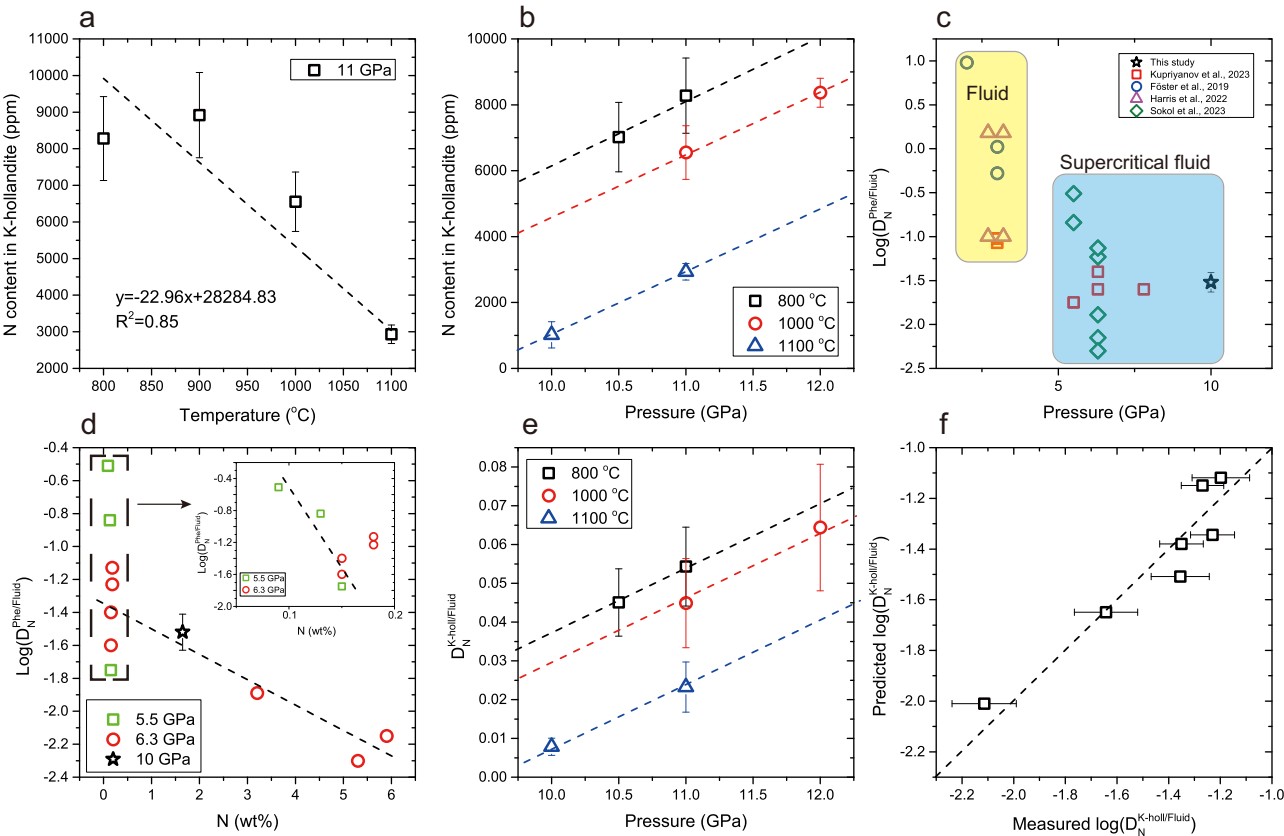

**Fig. 3 | Nitrogen contents and mineral-fluid nitrogen partition coefficients of phengite and K-hollandite. a** Temperature and (**b**) pressure dependence of the nitrogen content in K-hollandite. **c** Pressure effect on $\log(D_N^{Phe/Fluid})$. Yellow and blue areas represent nitrogen partition coefficients with aqueous fluid and supercritical fluid, respectively. **d** Nitrogen concentration effect on $\log(D_N^{Phe/Fluid})$. Data at 5.5 GPa and 6.3 GPa are from refs. 36,37. The slope of $\log(D_N^{Phe/Fluid})$ vs. $N$ (wt%) is much gentler at higher pressures. **e** Temperature and pressure effects on $D_N^{K-holl/Fluid}$. **f** Prediction performance of Eq. (2). Error bars represent the standard deviation. Source data are provided in Supplementary Data 1 and Table 1.

A fluid phase occurred in each of our experiments according to the sample observations (Supplementary Text 1) and mass balance calculations (Supplementary Text 2). To calculate the partition coefficients of nitrogen between phengite, K-hollandite, and fluid, we assumed that nitrogen was only distributed between the K-rich minerals (phengite and K-hollandite) and fluid due to the undetectable nitrogen in the other minerals. Previous authors usually assumed negligible silicate solutes[33,34] in the fluid when they calculated mineral-fluid partition coefficients ($D_N^{Mineral/Fluid}$). This assumption is appropriate for low-pressure fluids, while it may not be valid in this study because our fluids are supercritical according to previous studies on pelite-water systems[46]. It has been demonstrated that supercritical fluids can dissolve a significant amount of silicates[47,48]. To precisely estimate the contents of silicate solutes dissolved in our supercritical fluids, we used a machine learning approach (Methods). We developed two models that can respectively predict the silicate solubility in the aqueous fluid (fluid model) and the water solubility in hydrous melt (melt model) for systems with different compositions at various P-T conditions. The uncertainties of predictions for the two models are ~3.4 wt% and ~3.0 wt% (Supplementary Table 2). As the compositions of supercritical fluids should fall between those of aqueous fluids and hydrous melts[47], the predicted silicate solubility in the aqueous fluid and the silicate content in the hydrous melt (100 wt%−predicted water solubility) can be regarded as the lower and upper limits of the solute content dissolved in the supercritical fluid, respectively. The results show that our supercritical fluids dissolved ~19−55 wt% silicate, using which and nitrogen mass balance we can calculate the mineral-fluid partition coefficients of nitrogen, $D_N^{Phe/Fluid}$ and $D_N^{K-holl/Fluid}$ (Table 1).

The calculated $D_N^{Phe/Fluid}$ are about 0.031 ± 0.008 at 10 GPa and 1000 °C. Figure 3c shows our $D_N^{Phe/Fluid}$ are consistent with those obtained at pressures over 5.5 GPa[36,37] but lower than those obtained at lower pressures[27,33,36]. The difference may be due to the transition of aqueous fluid to supercritical fluid. Although the $NH_3/(NH_3 + N_2)$ ratio in fluid increases with increasing pressure[36,49], $D_N^{Phe/Fluid}$ seems to be independent on pressure. Rather, $D_N^{Phe/Fluid}$ decrease with increasing nitrogen concentration in the fluid[37]. However, we find that the effect of fluid nitrogen concentration on $D_N^{Phe/Fluid}$ may become weaker at high pressures (Fig. 3d).

The calculated $D_N^{K-holl/Fluid}$ vary with pressure and temperature, ranging from 0.008 ± 0.002 to 0.064 ± 0.016 (Fig. 3e). Linear regression of the experimental data yields the following equation to best describe $D_N^{K-holl/Fluid}$:

$$\log\left(D_N^{K-holl/Fluid}\right) = \frac{-2996.99(\pm 1365.49)}{T} + \frac{495.67(\pm 113.53)P}{T} - 3.44(\pm 0.51)(R^2 = 0.84, p-value = 0.01) \quad (2)$$

Temperature is in K and pressure is in GPa. The prediction performance is displayed in Fig. 3f. This equation implies that at the pressure range of phengite to K-hollandite transition increases with increasing pressure but decreases with increasing temperature. In combination with undetectable nitrogen in K-hollandite at oxidizing conditions, it can be concluded that during slab dehydration more nitrogen will be retained in K-hollandite in the reduced and cold slabs than in the oxidized and warm slabs.

## Assessment of equilibrium and redox conditions

All products have closely approached the equilibrium of nitrogen based on the following reasons: (1) Phengite and K-hollandite, as the main nitrogen-bearing minerals, have homogeneous nitrogen contents with relatively small standard deviations less than 15% (Supplementary Data 1). Besides, their homogeneous major element compositions show the same P-T dependence as those observed in previous studies with volatile-rich pelite (Fig. 2b, c). (2) For the runs ranging from 48 h to 72 h, the nitrogen content of K-hollandite shows a strong temperature dependence at 11 GPa and similar pressure dependences at different temperatures (800, 1000, and 1100 °C; Fig. 3a, b). (3) The variation of $D_N^{K-holl/Fluid}$ can be explained by the variation of the experimental P-T conditions (Fig. 3e; Eq. (2)).

Our experimental redox conditions were not controlled by any external oxygen fugacity buffers but should have been mainly determined by the used starting materials. We used the produced phase assemblages and compositions to approximately estimate the sample oxygen fugacity (Methods). The addition of $NH_4NO_3$, which decomposed into $N_2$, $H_2O$, and $O_2$ during the experiment, resulted in the presence of hematite in the $NH_4NO_3$-bearing run, which indicates that the oxygen fugacity of this run must have been above the $Fe_2O_3$-$Fe_3O_4$ (hematite-magnetite) buffer. The runs starting with $NH_3$ (25 wt%)-$H_2O$ solution must have had initial oxygen fugacity significantly below the Ni-NiO (NNO) buffer[49]. However, the absence of metal iron after the experiments indicates that the oxygen fugacity of these $NH_3$-bearing runs must have been higher than the Fe-FeO buffer. To further narrow down the oxygen fugacity estimates of our $NH_3$-bearing runs, we compared our experiments with the previous experiments conducted on pelite at 3–8 GPa by Sokol et al.[37], who also added reduced fluids ($C_3H_6N_6$) into Au capsules as the source of nitrogen. By analyzing the $NH_3/N_2$ ratios in their fluids after the experiments, Sokol et al.[37] estimated that the oxygen fugacity of their experiments starting with $C_3H_6N_6$ was about 3−4 log units below the NNO buffer. We also noticed that under subduction zone conditions, the FeO content of Cpx in pelite is positively correlated with oxygen fugacity (Methods; Supplementary Fig. 4). Using such a correlation we estimated that the oxygen fugacity of our $NH_3$-bearing runs should have been 2.7−3.1 log units below the NNO buffer. We therefore concluded that the oxygen fugacity of our $NH_3$-bearing runs should have been ~3 log units below the NNO buffer but above the Fe-FeO buffer. The oxygen fugacity of the subducted slabs maybe 0−2 log units below the NNO buffer[50,51], higher than the estimated oxygen fugacity of our $NH_3$-bearing runs. Therefore, considering the effect of oxygen fugacity on $D_N^{K-holl/Fluid}$, the actual $D_N^{K-holl/Fluid}$ could be even lower than the $D_N^{K-holl/Fluid}$ that were obtained from our $NH_3$-bearing runs and will be applied to the subduction zones. However, as shown below, lower $D_N^{K-holl/Fluid}$ would more support our conclusions on the deep subduction efficiency of slab sedimentary nitrogen.

## Deep subduction efficiency of slab sedimentary nitrogen

During the subduction of slab sediments, nitrogen is mainly stored in micas at shallow depths and then in phengite and K-hollandite at depths beyond the sub-arc depth. Thus, the contribution of sediment subduction to the deep nitrogen transport is mainly constrained by the stabilities of phengite and K-hollandite and their nitrogen partitioning behavior. Our study shows that phengite in the nitrogen-bearing system can be stable up to ~300−315 km (~10−10.5 GPa) (Fig. 2a), which is the maximum depth that nitrogen can be transported to by phengite. Once the transition from phengite to K-hollandite occurs at this depth, there should be a significant nitrogen loss because the $D_N^{K-holl/Fluid}$ are smaller than $D_N^{Phe/Fluid}$ (0.013 vs. 0.031) at the transition conditions, as calculated using Eq. (2). More importantly, a large amount of fluid is produced during the phengite to K-hollandite transition, which together with the low $D_N^{K-holl/Fluid}$ values (far below 1) suggests that significant nitrogen could be released into the mantle during the phengite

to K-hollandite transition. Therefore, the phengite to K-hollandite transition may play as an important filter for deep nitrogen subduction. To evaluate the fractions of nitrogen preserved during the phengite to K-hollandite transition, we take the Izu-Bonin-Mariana (IBM) and Central America (CA) subduction slabs as examples of cold and warm slabs, respectively[21,22]. Using our machine learning models and the phase proportions in this study (Methods), we estimate the mass ratios of K-hollandite to supercritical fluid during the phengite to K-hollandite transition, which is ~32 and ~28 for the cold IBM slab and the warm CA slab, respectively. We then use Eq. (2) and the P-T conditions of the phengite to K-hollandite transition[52] to calculate $D_N^{K-holl/Fluid}$, which are 0.024 for the cold IBM slab and 0.013 for the warm CA slab. Based on these data, the fractions of nitrogen preserved in K-hollandite are ~43.3% for the cold IBM slab and ~26.0% for the warm CA slab during the phengite to K-hollandite transition (Fig. 4a).

Knowing the fractions of nitrogen preserved, we can estimate the deep subduction efficiency of slab sedimentary nitrogen, which is the fraction of slab sedimentary nitrogen that can pass through the phengite to K-hollandite transition. A previous study showed that ~49−89% of sedimentary nitrogen may pass through the sub-arc depth for the cold IBM slab[21]. For the warm CA slab, we estimate that ~24−89% of sedimentary nitrogen may pass through the sub-arc depth, using available nitrogen influx and outflux data[22,28,29,53] (Supplementary Text 3). For simplification in our cases for the cold IBM slab and warm CA slab, we use average values (~69% and ~56%) for the fractions of slab sedimentary nitrogen that pass through the sub-arc depth, respectively. As a further step, assuming that after the sub-arc depth, phengite is the only hydrous mineral in slab sediments that carries water and nitrogen down to the phengite to K-hollandite transition depth and that its abundance does not significantly decrease along both cold and warm slab geothermal paths[46] (Fig. 4a), we can estimate that ~30% and ~15% of the initial sedimentary nitrogen can pass through the phengite to K-hollandite transition for the cold IBM slab and the warm CA slab, respectively.

However, it should be noted that the above-estimated nitrogen deep subduction efficiencies (~30% and ~15%) for IBM and CA slabs represent maximum values due to the following two reasons. First, as mentioned above, considering the effect of oxygen fugacity on $D_N^{K-holl/Fluid}$, the actual $D_N^{K-holl/Fluid}$ in subduction zones could be even lower than the applied $D_N^{K-holl/Fluid}$ based on our $NH_3$-bearing run with the oxygen fugacity of ~NNO-3. Second, any breakdown or dehydration of phengite at the depth before the phengite to K-hollandite transition[54] would decrease the estimated nitrogen deep subduction efficiency; and any reaction between phengite and fluids derived from the subducting slab, such as those from the serpentinite, would also decrease the estimated nitrogen deep subduction efficiency. For example, assuming a serpentinized slab mantle with a thickness of 2 km and water content of 2 wt%[55], complete dehydration of the serpentinized slab mantle at ~6 GPa in the warm CA slab would result in only ~7% of slab sedimentary nitrogen transported to the phengite to K-hollandite transition depth (Supplementary Fig. 5, Supplementary Data 2), and finally only ~2% of slab sedimentary nitrogen can be retained after the phengite to K-hollandite transition.

## Sedimentary nitrogen influx and origin of the lower mantle nitrogen

The global influx of slab sediments is $1.4 \times 10^{12}$ kg/y, including $5.3 \times 10^{11}$ kg/y from the cold subduction zones, $7.8 \times 10^{11}$ kg/y from the warm subduction zones, and $0.4 \times 10^{11}$ kg/y from the two hottest subduction zones (Mexico and Cascadia)[15,56]. Combined with the average nitrogen concentration of 560 ppm in the slab sediments[5], we calculated a nitrogen influx of slab sediments as ~$7.6 \times 10^{8}$ kg/y from the global slabs, $3.0 \times 10^{8}$ kg/y from the cold slabs, $4.4 \times 10^{8}$ kg/y from the warm slabs, and $0.2 \times 10^{8}$ kg/y from the hottest slabs. This bulk global sedimentary nitrogen influx is consistent with a previous estimate of

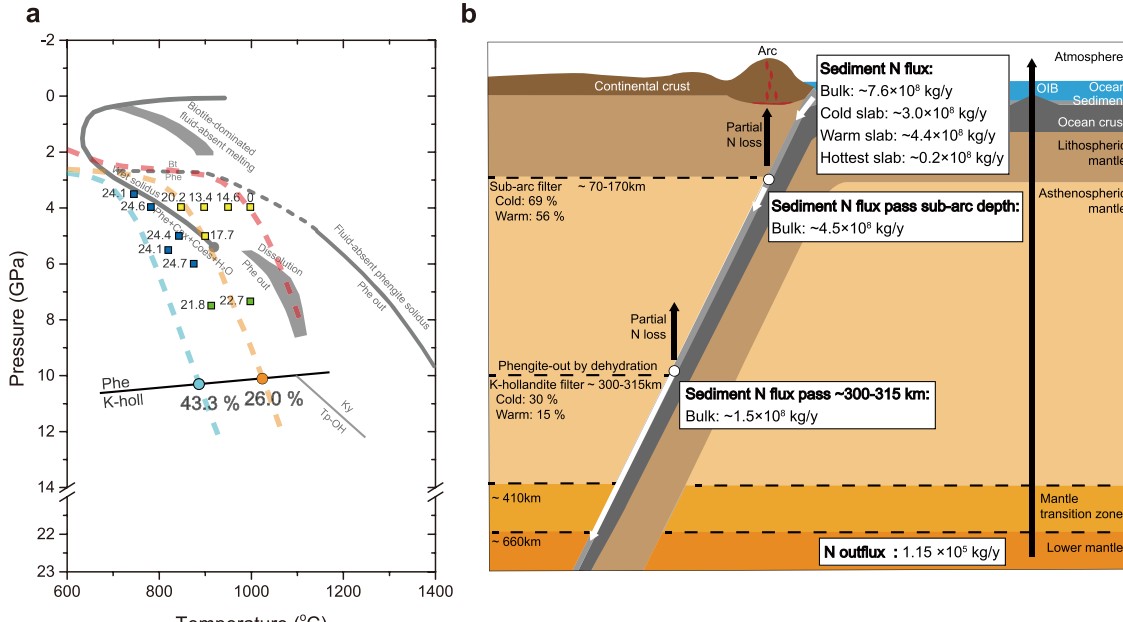

**Fig. 4 | Nitrogen subduction contributed by slab sediments. a** Phase diagram modified from Schmidt et al.[46], and the phengite to K-hollandite transition is based on Fig. 2a. Circle symbols indicate the phengite to K-hollandite transition where a large fraction of nitrogen is lost along the IBM slab (cyan dashed line) and the CA slab (orange dashed line); the numbers beside the circle symbols are the calculated fractions of nitrogen preserved. The P-T conditions for the slab surfaces (D80 model[52]) are extrapolated to 12 GPa. The red dashed line represents the geothermal gradient of the hottest slab (Mexico[52]), where no phengite exists below the sub-arc depth. The numbers beside the square symbols are the phase proportions (wt%) of phengite at the corresponding pressure and temperature[46]; the square symbols with blue color indicate sub-solidus runs, while the square symbols with yellow color mean that partial melting occurs, and the square symbols with green color indicate runs with supercritical fluids. **b** Schematic diagram showing the deep nitrogen flux of global sediments that pass through the sub-arc depth (~4.5 × 10$^8$ kg/y) and the phengite to K-hollandite transition depth (~1.5 × 10$^8$ kg/y). The nitrogen degassing flux (~1.15 × 10$^5$ kg/y) from the lower mantle is from ref. 85. See the main text for more details. Source data are provided in Supplementary Data 2.

~7.6 × 10$^8$ kg/y[23]. Using the deep subduction efficiencies of sedimentary nitrogen of the cold IBM slab and the warm CA slab, we can estimate the combined nitrogen influx of the global cold and warm slabs (here we ignore the two hottest slabs) that can pass through the sub-arc depth and the phengite to K-hollandite transition depth. The results (Fig. 4b) show that the sedimentary nitrogen influx that can pass through the sub-arc depth is ~4.5 × 10$^8$ kg/y, and the sedimentary nitrogen influx that can pass through the phengite to K-hollandite transition depth is ~1.5 × 10$^8$ kg/y. Again, it should be noted that 1.5 × 10$^8$ kg/y is the maximum sedimentary nitrogen influx, because we only use the maximum deep subduction efficiencies (~30% and ~15%) for the cold and warm slabs, respectively.

Assuming that plate tectonics started ~3 Ga ago[57] with a steady nitrogen net influx, and that all K-hollandite after the phengite to K-hollandite transition can be subducted into the lower mantle[38,58], we can estimate that in total ~4.6 × 10$^{17}$ kg nitrogen from slab sediments has been subducted to the lower mantle, which corresponds to ~0.15 ppm nitrogen in the lower mantle. This number could be significantly decreased if considering that the ancient slab geothermal could be warm to hot[59]. Using the N$_2$/Ar ratios in oceanic basalts and mantle xenoliths, 1–6 ppm nitrogen has been estimated for the Earth's mantle[5,60,61]. If we assume that the lower mantle has a nitrogen abundance also between 1–6 ppm, then only 2.5–15% of the lower mantle nitrogen is derived from slab sediments. A limited contribution of slab sediments to the lower mantle is actually consistent with two recent studies based on nitrogen isotopes, which conclude that the nitrogen in the mantle plume sources is a primordial feature[62,63]. Another recent study on the accretion of Earth's nitrogen demonstrates that the nitrogen budget and isotope composition of Earth's different reservoirs may have already been established during the main accretion phase[64], which is also consistent with our conclusion that the contribution of slab sediments to the lower mantle nitrogen is very limited.

## Methods

### Starting materials

The starting material powder had the same pelite composition as that used in Ono[39]. and was prepared from analytical reagent oxides (SiO$_2$, TiO$_2$, Al$_2$O$_3$, FeO, and MgO) and carbonates (CaCO$_3$, Na$_2$CO$_3$, and K$_2$CO$_3$). To remove absorbed water, SiO$_2$, TiO$_2$, Al$_2$O$_3$, and MgO powders were heated at 1000 °C for 12 h, Na$_2$CO$_3$, and K$_2$CO$_3$ powders were fired at 100 °C for 5 h, and CaCO$_3$ powder was held at 200 °C for 5 h. These chemical powders were mixed and ground in ethanol in an agate mortar for at least 1 h and subsequently dried at room temperature, after which the dried mixture was decarbonated at 1000 °C for 10 h. Finally, the FeO powder was mixed with the decarbonated mixture and ground in ethanol in an agate mortar for at least 1 h, after which the final mixture was dried at room temperature. All the dried starting materials were stored in a vacuum oven at 100 °C for at least 24 h before being loaded into the Au capsules. NH$_3$ (25 wt%)-H$_2$O solution and NH$_4$NO$_3$ were used as nitrogen sources for the eight NH$_3$-bearing runs and one NH$_4$NO$_3$-bearing run, respectively. For the NH$_3$-bearing runs, the starting material and the NH$_3$ solutions were enclosed in an Au capsule following a mass ratio of 92:8, which should generate a pelite system with 6 wt% H$_2$O and 2 wt% NH$_3$. For the NH$_4$NO$_3$-bearing run, NH$_4$NO$_3$ and H$_2$O provided 6 wt% H$_2$O and 1.65 wt% *N* for the system. The uncertainty induced by sample loading is discussed in Supplementary Text 4.

### Experimental procedures

High-pressure and high-temperature experiments were conducted at the Guangzhou Institute of Geochemistry, Chinese Academy of Sciences, using a 2500 ton Cubic-type multi-anvil apparatus (UHP-2500) with tungsten carbide cubes as secondary anvils. An 18/11 octahedral assembly that contained a Cr$_2$O$_3$-doped MgO octahedron and a LaCrO$_3$ heater was used. All the experiments were carried out at 10–12 GPa and

800–1100 °C for 48–72 h. The assemblies were initially pressured to target pressures within ~20 h, and then the temperature was increased at a rate of ~33 °C/min to the desired temperature. The temperatures were controlled using C-type ($W_{95}Re_5$-$W_{74}Re_{26}$) thermocouples, which were placed on the sample inside a 4-hole $Al_2O_3$ sleeve. The pressures were calibrated by phase transitions of Bi (I−II, 2.55 GPa), Bi (III−V, 7.7 GPa), Pb (13.6 GPa), and ZnS (15.6 GPa) at room temperature, and quartz-coesite (3.5 GPa), coesite-stishovite (9.9 GPa), and olivine-wadsleyite (14.5 GPa) at 1400 °C (Supplementary Fig. 6). The phase transitions used here are similar to that used in ref. [39], so uncertainty induced by pressure calibration should be small. Finally, all the samples were quenched to room temperature within a few seconds by switching off the electrical power, followed by ~20 h of decompression.

The chemical compositions of the quenched minerals were analyzed using an EPMA-1720 (Shimadzu) and a JEOL JXA-8230 Electron Probe Microanalyzer under operating conditions of 15 kV, 10–20 nA, and a 1–10 μm beam diameter. For phase identification and textural analysis, a Phenom XL scanning electron microscope with an energy-dispersive X-ray spectrometer and an HPRIBA LABRAM-HR Raman spectrometer with a 532 nm green line of a Spectra-Physics Ar laser as an excitation light source were used in this study.

### Nitrogen analysis of silicate minerals

Nitrogen concentrations were quantified using a JEOL JXA-8230 Electron Probe Microanalyzer equipped with five wavelength-dispersive X-ray spectrometers. An LDE1L diffracting crystal was used. Each sample was coated with a ~ 20 nm carbon film[65]. During the analysis of phengite and K-hollandite, an accelerating voltage of 10 kV, a beam current of 50 nA, and a spot size of 10 μm were applied. The counting times for the peak positions and background positions were both 100 s. To decrease the detection limit of garnet and clinopyroxene, the beam current and the counting time of the peak positions were changed to 200 nA and 200 s, respectively. The boron nitride (BN) was used as a standard sample. The $N$ Kα peak of BN was located at 147.54 ± 0.003 mm, and the $N$ Kα peak of K-hollandite was at 147.21 ± 0.01 mm (Supplementary Text 5; Supplementary Fig. 3). Because the background of nitrogen is a curved line rather than a straight line, the two-point background interpolation method is ineffective for subtracting the background and will cause significant uncertainty in nitrogen analysis. To solve this problem, we employed an exponential equation to subtract the $N$ background, which has previously been applied successfully in several studies[34,66–68]. The background line was obtained by fitting the exponential equation to four to five background points on either side of the $N$ Kα peak. A carbon film correction[69] was performed to prevent the effects due to the carbon film thickness difference between the sample and the standard. Furthermore, we performed ZAF correction and adjusted the peak overlap of Ti Ll on $N$ Kα. The detection limits of $N$ were ~ 700 ppm and ~800–1100 ppm by weight for phengite and K-hollandite, respectively, and ~200–300 ppm by weight for garnet and clinopyroxene. The accuracy of nitrogen measurement by this method was confirmed by measuring a hyalophane standard with similar compositions to our samples (1108 ± 171 ppm measured by this method versus 1200 ppm[70], 1130 ± 150 ppm[71], 900 ± 300 ppm[34], and 933 ppm[72]). The nitrogen species in the silicate minerals were investigated by a HPRIBA LABRAM-HR Raman spectrometer, and the Raman spectra of the nitrogen-bearing minerals were acquired from 4000 cm$^{-1}$ to 50 cm$^{-1}$ with an acquisition time of 2 × (20–100) s for each range.

### Assessment of redox conditions

The redox conditions of each run are approximately constrained by the phase assemblages and compositions. In the $NH_4NO_3$-bearing run, the decomposition reaction of $NH_4NO_3 = N_2 + 2H_2O + 1/2O_2$ should occur at 1000 °C and cause an oxidized condition. Indeed, the occurrence of hematite in this run indicates that the oxygen fugacity is at least higher than the $Fe_2O_3$–$Fe_3O_4$ (hematite-magnetite) buffer. Moreover, the abnormal chemical compositions of the minerals also suggest the extremely oxidized condition of this run. The garnet in this run lacks an almandine composition, while enrichments of iron in kyanite and clinopyroxene are observed (Supplementary Data 1). The composition of clinopyroxene is similar to jadeite ($NaAlSi_2O_6$), with a minor diopside composition ($CaMgSi_2O_6$), but has an obviously lower Al content. This indicates that iron is mainly $Fe^{3+}$, which replaces $Al^{3+}$. In contrast, in the $NH_3$-bearing runs, negligible amounts of Fe were detected in clinopyroxene and kyanite. The negligible amounts of Fe in clinopyroxene thus suggest that most of Fe is still ferrous iron. The negligible amounts of Fe in kyanite also suggest ferrous iron in the systems because Fe content in kyanite correlates with the ratio of $Fe^{3+}$ to $Fe^{2+}$ of the whole rock[73]. Using all the experiments on pelite at subduction zone conditions, with reported oxygen fugacity and Cpx composition[37], we found a positive correlation between the FeO content of Cpx and oxygen fugacity (Supplementary Fig. 4). Using the FeO contents of our Cpx, we estimated that the oxygen fugacity of our $NH_3$-bearing runs should be 2.7–3.1 log units below the NNO buffer.

### Machine learning process and performance of models

The dataset for the fluid model comprises 758 samples from 36 publications, including data on silicate solubility in aqueous fluid and supercritical fluid. The dataset used for the melt model comprises 976 samples from 44 publications, including data on water solubility in hydrous melt and supercritical fluid. The datasets are provided in Supplementary Data 3. The temperature, pressure, mole number of major elements (Si, Ti, Al, Fe, Mg, Ca, Na, K, and O), total electronegativity, and total ionization energy of the system, as well as the weighted electronegativity of each element at high pressure calculated by the empirical formula[74], were used as input data (X), while the silicate solubility in fluid or water solubility in melt was used as the output data (Y). The Random Forest[75], Extremely Randomized Trees (Extra-Trees)[76], and Extreme Gradient Boosting (XGBoost)[77] algorithms were used in this study (for descriptions of the algorithms' principles, see the Supplementary Text 6).

Before training the models, the whole dataset was divided into a training set (80%) and a test set (20%) by the stratified random sampling function in the Scikit-learning library[78]. The training set was used to train the model while the test set was used to estimate the generalization ability of the models. To obtain robust performance estimates for each model, Monte Carlo cross-validation was applied. The training set was split into a subset (80%) for training and a subset (20%) for validation. This process was repeated 500 times, and the average RMSEs on the 500 validation sets were used to estimate the goodness of fit of the models. The RMSE can be expressed as:

$$RMSE = \sqrt{\frac{\sum_{i=1}^{n}(y_i - \hat{y}_i)^2}{n}} \quad (3)$$

where $y_i$ denotes the real value and $\hat{y}_i$ is the predicted value. As the whole dataset is relatively small, the finally fluid and melt models made available to researchers were trained on all the data so that the algorithms learned as much data as possible.

The predicted values of the test set plot 1:1 to the measured values and the RMSEs of the test sets are slightly lower than the average RMSEs of the validation sets (Supplementary Fig. 7), which indicates that the three models are not overfitting and can provide accurate predictions on unknown samples. The average RMSEs of the validation sets and the RMSEs of the test sets were compared (Supplementary Table 2), demonstrating that Extra Trees shows advantages over Random Forest and XGBoost for both the fluid and melt models. Therefore, the predictions of Extra Trees model were used to calculate the partition coefficients.

**Estimating mass ratios of K-hollandite and supercritical fluid during the quantification of the amount of nitrogen carried away from the slab by supercritical fluid**

The mass ratios of K-hollandite and supercritical fluid were calculated based on the experimental results of this study combined with machine learning models. Considering that there were no obvious variations in the phase proportions of K-hollandite and topaz-OH under P-T conditions, average values of ~25.6 wt% and ~6.4 wt%, respectively, were applied. The total free water content at the phengite to K-hollandite transition depth was considered to be the amount of water released from phengite after complete breakdown minus that stored in newly formed topaz-OH, following the equation:

$$M_{Free}^{H_2O} = X_{Phe} C_{Phe}^{H_2O} - X_{Tp-OH} C_{Tp-OH}^{H_2O} \qquad (4)$$

where the $M$ is mass fraction, $X$ is the phase abundance, and $C$ is the water content in phases. The water contents in phengite and topaz-OH were set as 4 wt% and 10 wt%[39], respectively, based on the molecular formula. A phase proportion of ~29.9 wt% for phengite at 10 GPa and 1000 °C was used for both slabs, and the amount of water released from phengite was thus ~1.2 wt%. The total free water content is thus set to ~0.56 wt%. The Extra Trees models were used to predict the solute solubility in the supercritical fluids, and the average values of the predicted solute solubility are 31 wt% for the IBM slab and 39 wt% for the CA slab. Therefore, the mass ratios of K-hollandite to supercritical fluid are ~32 for the IBM slab and ~28 for the CA slab (Supplementary Data 2).

## Data availability

All data analyzed during this study are included with this published article and its Supplementary Information. The source data are also deposited in https://doi.org/10.5281/zenodo.13208779.

## Code availability

The code for machine learning training is provided in https://doi.org/10.5281/zenodo.10958032.

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

## Acknowledgements
This work is supported by the National Natural Science Foundation of China (42172041, 41972038). Y. Li received support from the National Science Fund for Distinguished Young Scholars (Grant No. 42225302).

## Author contributions
W.H.H: Formal analysis, Investigation, Writing—original draft, Writing—review & editing; Y.Y.: Supervision, Conceptualization, Resources, Methodology, Writing—original draft, Writing—review & editing, Funding acquisition; Y.L.: Conceptualization, Resources, Methodology, Writing—original draft, Writing—review & editing, Funding acquisition; Z.X., S.Y.Y., and S.B.G.: Resources, Methodology; Q.K.X.: Writing—review & editing.

## Competing interests
The authors declare no competing interests.
