## [Peer Review File · Nature Communications]

REVIEWER COMMENTS

Reviewer #1 (Remarks to the Author):

I have been asked to review the machine learning component of the manuscript. My review will focus purely on the use of Extra Trees to calculate solute content in supercritical fluid. I must confess that domain of application and the high pressure-high temp experiments are outside my areas of expertise, so I will be very focused on the machine learning aspect of this work.

The main manuscript has barely any description of the machine learning method beyond the stated predictions in table 1. It would be useful to have some details beyond just the final predicted values in the main manuscript rather than relegate all the methods used to predict solute content to the supplementary material.

Review Supplementary Text 3:

This section contains a good preliminary explanation of the machine learning process. While the authors have used this method in previous papers as highlighted by the references, it would be ideal to add a few lines explaining the Extra Trees Algorithm and how it works. Additionally, there is no mention of why Extra-Trees was selected as the best algorithm for this data over more popular algorithms like Random Forest (which works similar to Extra Trees), or XGboost etc. It would be good to show that the choice of algorithm was based on Extra-Trees performing better than many of the commonly used regression models.

Line 131: What was the reason to choose a 90/10 split for training and validation? It is more common to have between a 70/30 split to an 80/20 in cases of stratified random sampling. If there was a specific reason for this choice, authors should explain this choice.

Line 137: It is good to see the use of RMSE to evaluate the error of the extra trees model.

Line 142: R-squared isn't a very reliable method to measure the success of non-linear regression algorithms like Extra-Trees. Especially when comparing to the other state of the art models that may have used linear regression. I would instead recommend comparing success of the model by comparing RMSE values with other similar models.

Overall, it seems the machine learning component is mainly a step in their larger workflow and I hope my comments help add important information for readers and other researchers. But because most of my review focused on information in a specific section of the supplementary material, I do not feel comfortable making a recommendation to accept or reject the manuscript based on my review alone.

Reviewer #2 (Remarks to the Author):

The manuscript “Inefficient nitrogen transport to the lower mantle by sediment subduction” by Huang et al., presents the role of sediment subduction in the origins of deep Earth nitrogen and examines the behaviour of sedimentary nitrogen during the transition of phengite to K-hollandite under conditions of 10-12 GPa and temperatures ranging from 800-1100°C. The study investigates the nitrogen partitioning coefficients between K-hollandite and fluid (5 experiments), as well as phengite and fluid (1 experiment), revealing the trends with increasing pressure (and temperature). These findings offer new insights into how much nitrogen remains preserved at the phengite-hollandite transition depth and how much might be transported to the lower mantle by K-hollandite. However, the study may fall short on some fronts as detailed below. Most importantly, there are potential contradictions regarding the depths at which nitrogen-rich diamonds are found, especially in relation to the residual pressure of ice-VII inclusions in diamonds. While this research provides a new picture of phengite/K-hollandite transition in sediment subduction's contribution to mantle nitrogen, these weaknesses need further clarification.

Lines 159-162: The nitrogen content in K-Hollandite jumps from about 1000 ppm at 10 GPa to a high value of 8000 ppm at 11 GPa. That's quite the leap in just a small pressure window! I wonder if the lower 1000 ppm at 10 GPa is because it's near the Phe/K-Holl boundary? I'd be cautious about extending this trend to much higher pressures. And also, the K-hollandite with around 1000 ppm nitrogen was at the highest experimental temperature, while the one with higher nitrogen formed at a cooler temperature. Could temperature be playing a role here? Also since the change is so strong, it should be demonstrated with more experiments as it may be due to outliers.

Lines 169-172: But how much nitrogen actually gets to the depth where phengite transitions to K-hollandite? Given that $D_N(\text{mineral}/\text{fluid})$ is pretty low, at about 0.031 for phengite/fluid, and considering phengite's presence from the start of subduction to 10 GPa, it's possible most of the nitrogen is already out by 10 GPa. And with the slab heating up from the inside, there's likely a lot of fluid escaping from its core and moving up through the phengite-rich top layer.

Lines 196-199: Now, phengite interacts with fluid coming from within the slabs, especially as the serpentinized mantle rocks of the slab heat up. If this fluid isn't rich in nitrogen, phengite might lose its nitrogen to it, especially since $D_N(\text{phengite}/\text{fluid})$ is quite low at 0.031.

Lines 200-203: This rapid nitrogen drop might hinge on the lower $D_N(\text{K-hollandite}/\text{fluid})$, which seems to be increasing quickly within a tight pressure range of 1 GPa. This might be because it's close to the phengite/k-hollandite transition, as mentioned earlier in lines 159-162.

Lines 205-207: Unless it's reacting with the ultramafic surrounding mantle rocks, given that K-hollandite is significantly more siliceous.

Lines 281-287: This is based on the assumption that nitrogen degassing from the mantle (at 7.73×10^7 kg/y) has been consistent throughout Earth's history. But, with Earth cooling down, there's been a

decline in magmatism, as seen in the slowing rates of MORB spreading, reduced partial melting, and cooler upper mantle temperatures. So, it's likely that nitrogen degassing has been decreasing over time, which might make these calculations a bit off.

Lines 291-294: Some earlier research suggests that Earth's ancient atmosphere had a pressure pretty similar to what we have today. Just check out the studies by Som and colleagues in 2013 (Nature) and 2016 (Nature Geoscience), and the one by Marty and team in 2013 (Science).

Lines 307-310: It's not too surprising that nitrogen-rich diamonds are mainly found at shallower depths. After all, the subduction of nitrogen-rich materials tends to happen from the top down.

Lines 313-316: The Science study you cited, titled "Ice-VII inclusions in diamonds: Evidence for aqueous fluid in Earth's deep mantle", shows the current residual pressure of the ice-VII inclusions at 9 ± 2 GPa. However, the diamonds containing these inclusions actually formed much deeper, around the mantle transition zone (14-22 GPa and even beyond 22 GPa). This seems a bit contradictory to your results, especially since this study's main takeaway is that diamonds primarily form at around 310-315 km or roughly 10 GPa (as shown in Fig. 4). At 9 ± 2 GPa, ice-VII can only exist below 500 C, but even the coldest slabs are hotter than 800 C at this pressure. This might need some rethinking or further clarification.

Reviewer #3 (Remarks to the Author):

Huang et al present new experimental data to constrain the partitioning of N during the mineralogical transition of phengite and K-hollandite. They find that N partitioning is weaker into K-hollandite than into phengite, and this implies that slabs could lose a large fraction of remaining N budget as they pass through 10 GPa (~300 km depth), potentially limiting the supply of N to deeper sections of the mantle.

While the essential data reported may provide an important constraint regarding how slabs work to redistribute N in Earth's interior, the manuscript needs to include a section that discusses evidence for equilibrium being closely approached for the experiments, with careful attention paid to N. Without this, N partition coefficients cannot be established, as partitioning is a reflection of an equilibrium chemical system.

Moreover, previous works have identified that oxygen fugacity is a major control on N partitioning, but this parameter is also not discussed. I think it is simply assumed that all N remains reduced, but this is not stated nor defended.

The mass balance calculations appear to assume that exactly 8 wt % of a fluid was added to oxide powders to generate the starting composition. It is not possible to add exactly 8 wt % of fluid to a capsule and then have 100 % retention with welding, and given the centrality of this number, discussion of the precision and accuracy of this number needs to be included.

The manuscript states that carbonates were used as sources of Ca, Na, and K. Typically these are fired in

air to decarbonate, but firing will oxidize the FeO (potentially leading to oxidation of N in experiments). If not fired, then clearly CO₂ will be a part of the system. More detail regarding starting composition preparation needs to be included. Was a LOI taken for the starting oxide composition to verify that it adds negligible water or CO₂?

I was lost in the section regarding diamonds and suggest it be removed to provide more focus on the essential data. Similarly, I had a hard time understanding the uncertainties regarding the N flux calculations. More robust number to focus on may be simply that fraction of N retained in the slab to the point of phengite breakdown is released.

I commend the authors for their attempt to calculate the mass and composition of the supercritical fluid in their experiments, but insufficient details were given for the reader to evaluate this work in the manuscript. It may be helpful to publish this work as a separate manuscript to provide space and focus for this work to be understood and evaluated.

The comparison with the Ono phengite-out determination is important, but the conclusion that N (at relatively low concentrations) can shift this boundary by ~2 GPa is weak. A comparison of bulk system chemistries and experimental approaches need to be considered. What is the pressure calibration of the presses and assemblies used in these studies? A more firm conclusion could be derived if the same starting composition and same press+assembly was used for N-bearing and N-free experiments.

Please discuss the choice of functional form used for the Eq on line 163 (and I presume that R₂ should be 0.92 not 92).

An important possibility to consider is that AOC and sediments may have water-rich fluid pass through them relating to the breakdown of serpentinites. The analysis of this paper appears to neglect this possibility.

Please discuss the K-alpha peak position of N in BN vs that in your samples.

A response to reviewers

Reply to the comments from Reviewer #1:

I have been asked to review the machine learning component of the manuscript. My review will focus purely on the use of Extra Trees to calculate solute content in supercritical fluid. I must confess that domain of application and the high pressure-high temp experiments are outside my areas of expertise, so I will be very focused on the machine learning aspect of this work.

The main manuscript has barely any description of the machine learning method beyond the stated predictions in table 1. It would be useful to have some details beyond just the final predicted values in the main manuscript rather than relegate all the methods used to predict solute content to the supplementary material.

Reply: Thanks for your suggestions, we have moved the descriptions of machine learning method from supplementary materials to the main manuscript, and the methods used to predict solute content were detailed in this revised main manuscript (**Lines 138-147; Methods lines 371-402**).

Review Supplementary Text 3:

This section contains a good preliminary explanation of the machine learning process. While the authors have used this method in previous papers as highlighted by the references, it would be ideal to add a few lines explaining the Extra Trees Algorithm and how it works. Additionally, there is no mention of why Extra-Trees was selected as the best algorithm for this data over more popular algorithms like Random Forest (which works similar to Extra Trees), or XGboost etc. It would be good to show that the choice of algorithm was based on Extra-Trees performing better than many of the commonly used regression models.

Reply: We have added the description about the principles of the Extra-Trees, Random Forest, and XGBoost algorithms in **Supplementary Text 5**. We have compared the performances of these three algorithms in this revision (**Methods lines 381-402; Supplementary Fig. 6; Supplementary Table 3**).

The Extra-Trees performs best with the lowest RMSEs for both fluid model and

melt model (Table R1-1). This result is in good agreement with several recent studies¹⁻³. The performance of the Extra-Trees is always a bit better than that of the Random Forest and XGBoost when solving regression problems with small data sets. It is possibly due to the Extra-Trees reduces overfitting through randomness, while for small data sets, overfitting is one of the most important factors that affect the performance of model.

Table R1-1 Performances of machine learning models.

	Extra Trees	Random Forest	XGBoost
Performance of fluid models			
RMSE (validation)	3.39 (101)	4.32 (112)	3.72 (111)
R ² (validation)	0.93 (4)	0.88 (5)	0.91 (5)
RMSE (test)	2.83	2.99	3.02
R ² (test)	0.95	0.94	0.94
Performance of melt models			
RMSE (validation)	3.04 (87)	3.84 (106)	3.32 (80)
R ² (validation)	0.98 (2)	0.96 (2)	0.97 (2)
RMSE (test)	2.10	2.78	2.41
R ² (test)	0.99	0.98	0.98

Line 131: What was the reason to choose a 90/10 split for training and validation? It is more common to have between a 70/30 split to an 80/20 in cases of stratified random sampling. If there was a specific reason for this choice, authors should explain this choice.

Reply: Thanks for your suggestion. We have modified and chosen an 80/20 split for training and test set, and explained in this revised manuscript (**Methods lines 384-392**).

The stratified random sampling function was applied for each split of data set, and an 80/20 split has been chosen for training and test set. The training set was used to train the model while the test set was used to estimate the generalization ability of models. To obtain robust performance estimate for each model, training set was repeatedly split into a subset (80%) for training and a subset (20%) for validating. This process was repeated for 500 times, and the average RMSEs of the 500 validation sets were used to estimate the performance of models.

Line 137: It is good to see the use of RMSE to evaluate the error of the extra trees model.

Reply: Thanks for the comment.

Line 142: R-squared isn't a very reliable method to measure the success of non-linear regression algorithms like Extra-Trees. Especially when comparing to the other state of the art models that may have used linear regression. I would instead recommend comparing success of the model by comparing RMSE values with other similar models.

Reply: Thanks very much for your suggestion. We have compared the performance of different models based on RMSEs rather than R^2 in this revision (Methods lines 395-402; Supplementary Fig. 6; Supplementary Table 3).

Overall, it seems the machine learning component is mainly a step in their larger workflow and I hope my comments help add important information for readers and other researchers. But because most of my review focused on information in a specific section of the supplementary material, I do not feel comfortable making a recommendation to accept or reject the manuscript based on my review alone.

Reply: Thanks very much for your suggestions. We believe the revised manuscript has been much improved.

Reply to the comments from Reviewer #2:

The manuscript “Inefficient nitrogen transport to the lower mantle by sediment subduction” by Huang et al., presents the role of sediment subduction in the origins of deep Earth nitrogen and examines the behaviour of sedimentary nitrogen during the transition of phengite to K-hollandite under conditions of 10-12 GPa and temperatures ranging from 800-1100°C. The study investigates the nitrogen partitioning coefficients between K-hollandite and fluid (5 experiments), as well as phengite and fluid (1 experiment), revealing the trends with increasing pressure (and temperature). These findings offer new insights into how much nitrogen remains preserved at the phengite-hollandite transition depth and how much might be transported to the lower mantle by K-hollandite. However, the study may fall short on some fronts as detailed below. Most importantly, there are potential contradictions regarding the depths at which nitrogen-rich diamonds are found, especially in relation to the residual pressure of ice-VII inclusions in diamonds. While this research provides a new picture of phengite/K-hollandite transition in sediment subduction's contribution to mantle nitrogen, these weaknesses need further clarification.

Reply: Thanks very much for your nice comments. Aimed at these shortcomings,

we have added some experiments, deepened the discussion and revised the manuscript point-by-point. We have deleted the discussion of the formation of diamond according to the suggestion of Reviewer #3.

Lines 159-162: The nitrogen content in K-Hollandite jumps from about 1000 ppm at 10 GPa to a high value of 8000 ppm at 11 GPa. That's quite the leap in just a small pressure window! I wonder if the lower 1000 ppm at 10 GPa is because it's near the Phe/K-Holl boundary? I'd be cautious about extending this trend to much higher pressures. And also, the K-hollandite with around 1000 ppm nitrogen was at the highest experimental temperature, while the one with higher nitrogen formed at a cooler temperature. Could temperature be playing a role here? Also since the change is so strong, it should be demonstrated with more experiments as it may be due to outliers.

Reply: Thanks very much for the suggestion. In this revision, we have added two new experiments at 11 GPa-1000 °C and 11 GPa-1100 °C to investigate the role of temperature in nitrogen content in K-hollandite. We have demonstrated the P-T dependence of nitrogen content in K-hollandite in the revised manuscript (**Lines 114-120**).

As shown in the following Fig. R2-1a, the nitrogen content of K-hollandite indeed depends on temperature and shows a decrease with increasing temperature at 11 GPa. Besides, at each investigated temperature, it increases with enhancing pressure following the same slope (Fig. R2-1b). Linear regression of the experimental data yields the following equation to predict N content in K-hollandite:

$$\log(C_N^{\text{K-holl}}, \text{ppm}) = \frac{-2481.25(\pm 1539.39)}{T} + \frac{482.05(\pm 127.99)P}{T} + 1.42(\pm 0.57)$$

($R^2=0.82$, p-value=0.01) (Eq. 1)

Therefore, the excellent P-T dependence can exclude the possible effect of Phe/K-holl boundary. Based on Eq. 1, the high values of ~8000-9000 ppm at 11 GPa-800/900 °C, and low value of ~1000 ppm at 10 GPa-1100 °C are expected. Thus, the drastic decrease of nitrogen content in K-hollandite of the 10 GPa-1100 °C run should be caused by the lower pressure and higher temperature.

Fig. R2-1 (a) The temperature and (b) pressure dependence of the nitrogen content in K-hollandite.

Lines 169-172: But how much nitrogen actually gets to the depth where phengite transitions to K-hollandite? Given that $D_N(\text{mineral/fluid})$ is pretty low, at about 0.031 for phengite/fluid, and considering phengite's presence from the start of subduction to 10 GPa, it's possible most of the nitrogen is already out by 10 GPa. And with the slab heating up from the inside, there's likely a lot of fluid escaping from its core and moving up through the phengite-rich top layer.

Reply: Thanks for pointing out this. We have modified this description in this revised manuscript (**Lines 163-168**).

Moreover, we have revised the discussion of nitrogen preservation in phengite and sedimentary nitrogen subduction efficiency passing through the phengite to K-hollandite transition in this revision (**Lines 203-239; Supplementary Table 4; Supplementary Fig. 3**).

First, we give a max estimate assuming there is no fluid provided by other sources. Previous study showed that ~49-89% of sedimentary nitrogen may pass through the sub-arc mantle for the cold IBM slab⁴. For the warm CA slab, we estimate that ~24-89% of sedimentary nitrogen may pass through the sub-arc depth, using available nitrogen influx and outflux⁵⁻⁸ (**Supplementary Text 3**). For simplification in our cases for the cold IBM slab and warm CA slab, we use average values (~69% and ~56%) for the fractions of slab sedimentary nitrogen passing through the sub-arc depth, respectively. Assuming that after the sub-arc depth phengite is the only hydrous mineral in slab sediments that carries water and nitrogen down to the phengite to K-hollandite

transition depth, and that its abundance does not significantly decrease along both cold and warm slab geothermal paths⁹, we can estimate that ~30% and ~15% of the initial sedimentary nitrogen can pass through the phengite to K-hollandite transition for the cold IBM slab and the warm CA slab, respectively. It should be noted that these should be maximum values. Any breakdown or dehydration of phengite at the depth before the phengite to K-hollandite transition¹⁰ would decrease the estimated nitrogen deep subduction efficiency; and any reaction between phengite and fluids derived from the subducting slab, such as those from the serpentinite, would also decrease the estimated nitrogen deep subduction efficiency.

Then, we give an estimate considering fluid supplied from serpentinitized slab mantle (Fig. R2-2). The amounts of fluid produced from serpentine dehydration depend on geotherms¹¹⁻¹³. We take the cold Izu-Bonin-Mariana (IBM) slab and warm Central America (CA) slab as representative slabs in our discussion. In the cold IBM slab, serpentine-bearing peridotite transfers to the Phase A-bearing peridotite and almost no fluid released¹¹, so that 100% nitrogen can be preserved in phengite and carried to the K-hollandite filter. On the other hand, serpentinitized slab mantle completely dehydrates at ~6 GPa in the warm CA slab, which will cause a significant nitrogen loss from phengite. In this scenario, assuming a serpentinitized slab mantle with a thickness of 2 km and water content of 2 wt%¹⁴, the nitrogen preservation in phengite is calculated to be only ~12.3% based on the reported value of $D_N^{\text{Phe/Fluid}}$ at 5.5 and 6.3 GPa^{15,16}. This results in only ~7% of slab sedimentary nitrogen transported to the phengite to K-hollandite transition depth, and only ~2% of slab sedimentary nitrogen passing through the phengite to K-hollandite transition in the CA slab. If with fluid from other sources poor in nitrogen, such as the slab oceanic crust, the nitrogen preservation in phengite to the K-hollandite filter of both cold and warm slabs can be even smaller, especially considering that $D_N^{\text{Phe/Fluid}}$ at 10 GPa is much smaller.

In short, if considering fluid supplied from other sources, sedimentary nitrogen subduction to the lower mantle would be more inefficient, further strengthening our conclusion.

Fig. R2-2 Illustration of nitrogen preservation of phengite considering serpentinite dehydration. P-T path of the slab Moho is from D80 model ¹². The phase relations of water saturated peridotite are displayed by black dashed lines ¹¹. In the cold slab (Izu-Bonin-Mariana, cyan line), antigorite directly transfers to Phase A and no fluid will be released up to the depth of mantle transition zone ¹¹. In the warm slab (Central American, orange line), antigorite completely dehydrates and significant fluid will be produced, which will cause nitrogen loss from phengite. 100% and 12.3% are the estimated nitrogen preservations in phengite of the cold slab and warm slab, respectively.

Lines 196-199: Now, phengite interacts with fluid coming from within the slabs, especially as the serpentinitized mantle rocks of the slab heat up. If this fluid isn't rich in nitrogen, phengite might lose its nitrogen to it, especially since $DN(\text{phengite}/\text{fluid})$ is quite low at 0.031.

Reply: Yes. We have discussed the nitrogen preservation in phengite considering fluid existing, and thereby revised the discussion about sedimentary nitrogen subduction efficiency passing through the phengite to K-hollandite transition in this revision (Lines 228-239; Supplementary Table 4; Supplementary Fig. 3).

As the response above, without considering fluid supplied from other sources, we estimate the maximum values of the initial sedimentary nitrogen that can pass through the phengite to K-hollandite transition for the cold IBM slab of 30% and the warm CA slab of 15%. It should be noted that any breakdown or dehydration of phengite at the depth before the phengite to K-hollandite transition ¹⁰ would decrease the estimated nitrogen deep subduction efficiency; and any reaction between phengite and fluids derived from the subducting slab, such as those from the serpentinite, would also decrease the estimated nitrogen deep subduction efficiency. It depends on the amount of fluid in the slabs, especially supplied by serpentine dehydration subject to geotherms ^{11–13}. We take the cold Izu-Bonin-Mariana (IBM) slab and warm Central America (CA) slab as representative slabs in our discussion. In the cold IBM slab, serpentine-bearing peridotite transfers to the Phase A-bearing peridotite and almost no fluid released ¹¹, so that 100% nitrogen can be preserved in phengite and carried to the K-hollandite filter. On the other hand, serpentinitized slab mantle completely dehydrates at ~6 GPa in the warm CA slab, which will cause a significant nitrogen loss from phengite. In this scenario, assuming a serpentinitized slab mantle with a thickness of 2 km and water content of 2 wt% ¹⁴, the nitrogen preservation in phengite is calculated to be only ~12.3% based on the reported value of $D_N^{\text{Phe/Fluid}}$ at 5.5 and 6.3 GPa ^{15,16}. This results in only ~7% of slab sedimentary nitrogen transported to the phengite to K-hollandite transition depth, and only ~2% of slab sedimentary nitrogen passing through the phengite to K-hollandite transition in the CA slab. If with fluid from other sources poor in nitrogen, such as the slab oceanic crust, the nitrogen preservation in phengite to the K-hollandite filter of both cold and warm slabs can be even smaller, especially considering that $D_N^{\text{Phe/Fluid}}$ at 10 GPa is much smaller.

In short, if only considering fluid poor in nitrogen supplied from serpentine dehydration, nitrogen preservation in phengite from the sub-arc depth to the phengite to K-hollandite transition depth is estimated to be 100 % for cold slab, while 12.3% for the warm slab. Accordingly, sedimentary nitrogen subduction efficiency passing through the phengite to K-hollandite transition depth is ~30% for the cold slab, while ~2% for the warm slab. Obviously, sedimentary nitrogen subduction to the lower mantle would be more inefficient if considering fluid from the serpentinitized slab mantle.

Lines 200-203: This rapid nitrogen drop might hinge on the lower $D_N(\text{K-hollandite/fluid})$, which seems to be increasing quickly within a tight pressure range of 1 GPa. This might be because it's close to the phengite/k-hollandite transition, as mentioned earlier in lines 159-162.

Reply: It is mainly caused by the temperature effect, which is demonstrated by the data from our added experiments. We have discussed the P-T dependence of $D_N^{\text{K-holl/Fluid}}$ in this revised manuscript (**Lines 157-165**). We have also modified the comparison between $D_N^{\text{Phe/Fluid}}$ and $D_N^{\text{K-holl/Fluid}}$ at the same P-T condition, 0.031 versus 0.013 at 10 GPa-1000 °C (**Lines 196-198**).

As the above response, nitrogen content in K-hollandite increases with pressure but decreases with temperature, which results in the drop of $D_N^{\text{K-holl/Fluid}}$ at 10 GPa-1100 °C compared with $D_N^{\text{Phe/Fluid}}$ at 10 GPa-1000 °C. $D_N^{\text{K-holl/Fluid}}$ is thus strongly controlled by temperature and pressure (Fig. R2-3), which can be described as the following equation:

$$\log(D_N^{\text{K-holl/Fluid}}) = \frac{-2996.99(\pm 1365.49)}{T} + \frac{495.67(\pm 113.53)P}{T} - 3.44(\pm 0.51)$$

($R^2=0.84$, p-value=0.01) (Eq. 2)

Where temperature is in K, and pressure is in GPa.

Fig. R2-3 (a) Temperature and pressure effects on the $D_N^{\text{K-holl/Fluid}}$. (b) Prediction performance of Eq. 2

Lines 205-207: Unless it's reacting with the ultramafic surrounding mantle rocks, given that K-hollandite is significantly more siliceous.

Reply: Thanks for pointing out this. In order to evaluate the contribution of sedimentary nitrogen to the origin of nitrogen in the lower mantle, we have estimated a maximum nitrogen flux carried by K-hollandite to the lower mantle, assuming that all K-hollandite after the phengite to K-hollandite transition can be subducted into the lower mantle in the revised manuscript (**Lines 259-260**).

We agree that K-hollandite is more siliceous and may react with ultramafic surrounding mantle rocks. But recent studies suggest that the composition of mantle is inhomogeneous and may become more silica-rich than pyrolite in the deep mantle¹⁷⁻¹⁹. Notably, K-hollandite inclusions have been observed in some diamonds from the lower mantle²⁰⁻²². These evidences suggest that K-hollandite is possible to exist in the deep mantle, depending on the model mantle composition providing sufficient K, Si, and Al. In our study, in order to evaluate the contribution of sedimentary nitrogen to the origin of nitrogen in the lower mantle, we just estimate a maximum nitrogen flux carried by K-hollandite, assuming that K-hollandite does not react with the surrounding mantle rocks in the lower part of upper mantle and the mantle transition zone.

Lines 281-287: This is based on the assumption that nitrogen degassing from the mantle (at 7.73×10^7 kg/y) has been consistent throughout Earth's history. But, with Earth cooling down, there's been a decline in magmatism, as seen in the slowing rates of MORB spreading, reduced partial melting, and cooler upper mantle temperatures. So, it's likely that nitrogen degassing has been decreasing over time, which might make these calculations a bit off.

Reply: Yes. With Earth cooling down, it's likely that nitrogen degassing has been decreasing over time. But we here provide a maximum estimation of nitrogen accumulation to the lower mantle applying the current values of influx and degassing flux, in order to evaluate the contribution of sedimentary nitrogen to the origin of the lower mantle nitrogen. We have stated this in this revised manuscript (**Lines 258-263**).

Assuming that plate tectonics started ~ 3 Ga ago²³ with a steady nitrogen net influx, and all K-hollandite after the phengite to K-hollandite transition can be subducted into the lower mantle^{21,24}, we can estimate that $\sim 4.6 \times 10^{17}$ kg nitrogen from slab sediments has been subducted to the lower mantle, which corresponds to ~ 0.15 ppm nitrogen in

the lower mantle. The estimated nitrogen amount to the lower mantle should be a maximum value, because the ancient slab geothermal could be warm to hot ²⁵, which could readily increase outgassing and decrease the net nitrogen influx by slab sediments to the lower mantle.

Based on this maximum estimation, sedimentary nitrogen contributes 2.5-15% nitrogen in the present lower mantle. As a result, the origin of nitrogen in the lower mantle from sedimentary nitrogen should be very limited, less than ~15% considering the higher outgassing flux in the past.

Lines 291-294: Some earlier research suggests that Earth's ancient atmosphere had a pressure pretty similar to what we have today. Just check out the studies by Som and colleagues in 2013 (Nature) and 2016 (Nature Geoscience), and the one by Marty and team in 2013 (Science).

Reply: Thanks for your suggestion. We have read studies by Som and colleagues in 2013 (Nature) and 2016 (Nature Geoscience), and the one by Marty and team in 2013 (Science). We noted that Som et al. ^{26,27} and Marty et al. ²⁸ provided nitrogen partial pressure of Archean atmosphere at 2.7-3.5 Ga ago. But Yoshioka et al. ²⁹ suggested the nitrogen partial pressures of a primordial atmosphere coexisting with the crystallizing magma ocean.

To evaluate the contribution of sedimentary nitrogen to the origin of the lower mantle nitrogen, we do not apply the initial mantle nitrogen budget of Yoshioka et al. ²⁹, but have chosen the present mantle nitrogen budget obtained from N₂/Ar ratios ³⁰⁻³² in this revised manuscript (**Lines 263-267**).

Lines 307-310: It's not too surprising that nitrogen-rich diamonds are mainly found at shallower depths. After all, the subduction of nitrogen-rich materials tends to happen from the top down.

Reply: Yes. We have deleted the discussion of formation of diamond in this revised manuscript.

Lines 313-316: The Science study you cited, titled "Ice-VII inclusions in diamonds: Evidence for aqueous fluid in Earth's deep mantle", shows the current residual pressure of the ice-VII inclusions at 9 +/- 2 GPa. However, the diamonds containing these inclusions actually formed much deeper, around the mantle transition zone (14-22 GPa

and even beyond 22 GPa). This seems a bit contradictory to your results, especially since this study's main takeaway is that diamonds primarily form at around 310-315 km or roughly 10 GPa (as shown in Fig. 4). At 9 +/- 2 GPa, ice-VII can only exist below 500 C, but even the coldest slabs are hotter than 800 C at this pressure. This might need some rethinking or further clarification.

Reply: Yes. We have removed this part and mainly paid attention to the contribution of sedimentary nitrogen to the origin of nitrogen in the lower mantle, following the suggestion of reviewer #3.

Reply to the comments from Reviewer #3:

Huang et al present new experimental data to constrain the partitioning of N during the mineralogical transition of phengite and K-hollandite. They find that N partitioning is weaker into K-hollandite than into phengite, and this implies that slabs could lose a large fraction of remaining N budget as they pass through 10 GPa (~300 km depth), potentially limiting the supply of N to deeper sections of the mantle.

While the essential data reported may provide an important constraint regarding how slabs work to redistribute N in Earth's interior, the manuscript needs to include a section that discusses evidence for equilibrium being closely approached for the experiments, with careful attention paid to N. Without this, N partition coefficients cannot be established, as partitioning is a reflection of an equilibrium chemical system.

Reply: Yes, thanks very much for the nice comment and suggestion. To evaluate equilibrium, we have added two experiments for different durations, i.e., 48 h and 72 h. Based on the homogenous distribution of nitrogen and major elements, and excellent P-T dependence of nitrogen content of K-hollandite and $D_N^{K-holl/Fluid}$, we have added a section to discuss the attainment of equilibrium in this revised manuscript (**Lines 170-179**).

All products have closely approached the equilibrium of nitrogen based on the following reasons: (1) As the only nitrogen-bearing minerals in the run products, phengite and K-hollandite of each run have homogeneous nitrogen content with relatively small standard deviations less than 15%. Besides, their homogeneous major element compositions display the same P-T dependence to those observed in previous studies with volatile-rich pelite (Fig. R3-1a, b). (2) For the runs ranging from 48 h to

72 h, the nitrogen content of K-hollandite shows a strong temperature dependence at 11 GPa (Fig. R3-1c) and similar pressure dependences at different temperatures (800, 1000 and 1100 °C; Fig. R3-1d). (3) The variation of $D_N^{K\text{-holl}/\text{Fluid}}$ can also be explained by the variation of the experimental P-T conditions (Fig. R3-1e).

Fig. R3-1 (a) Pressure dependence of major elements in phengite. (b) Temperature dependence of major elements in K-hollandite. (c) Temperature dependence of nitrogen content in K-hollandite. (d) Pressure and temperature dependences of nitrogen content in K-hollandite. (e) Pressure and Temperature effects on $D_N^{K\text{-holl}/\text{Fluid}}$.

Moreover, previous works have identified that oxygen fugacity is a major control on N partitioning, but this parameter is also not discussed. I think it is simply assumed that all N remains reduced, but this is not stated nor defended.

Reply: Thanks for pointing out this. Oxygen fugacity should be an important parameter that influence nitrogen partitioning. To evaluate potential impact of oxygen fugacity on the nitrogen partitioning behavior in our study, we have added two experiments at 11 GPa and 1000 °C with NH_3 solution and NH_4NO_3 as the source of nitrogen, respectively. We have evaluated the oxygen fugacity of our experimental systems in this revised manuscript (Lines 165-168; lines 180-187; Methods lines 347-370).

(1) Oxygen fugacity evaluation. The oxygen fugacity in our experiments is not

controlled by buffer, but it can be roughly constrained by the phase assemblages and the compositions of phases. In the NH_4NO_3 -bearing run, the decomposition reaction of $\text{NH}_4\text{NO}_3 = \text{N}_2 + 2\text{H}_2\text{O} + 1/2\text{O}_2$ should occur at 1000 °C and cause an oxidized condition. Indeed, the observation of hematite in this run indicates that the oxygen fugacity is at least higher than $\text{Fe}_2\text{O}_3\text{--Fe}_3\text{O}_4$ (HM) buffer. As shown in the following Fig. R3-2, the oxidized condition is also evidenced by the abnormal mineral compositions. Garnet in this run is lack of almandine composition (Fig. R3-2a), while the enrichments of iron in kyanite and clinopyroxene are observed. The composition of clinopyroxene is similar to jadeite ($\text{NaAlSi}_2\text{O}_6$), with a minor diopside composition ($\text{CaMgSi}_2\text{O}_6$), but has obvious lower Al content. It indicates that iron is mainly Fe^{3+} which replaces the Al^{3+} (Fig. R3-2b). On the contrary, in the NH_3 -bearing runs, negligible iron is detected in clinopyroxene and kyanite, suggesting that most of Fe is still ferrous iron. Clinopyroxene formed, at 10-12 GPa and 1000-1100 °C, in a Ni-NiO (NNO) buffered pelite system with ~4.12 wt% iron contains ~1.18-1.68 wt% iron³³, much higher than the Fe content of our clinopyroxene formed in a pelite system with ~7 wt% iron. This comparison suggests that the oxygen fugacity of the NH_3 -bearing runs should be at least lower than NNO buffer. On the other hand, the absence of metal iron indicates that the oxygen fugacity of the NH_3 -bearing runs is higher than Fe-FeO (IW) buffer. Therefore, the oxygen fugacity of the NH_3 -bearing runs can be limited between NNO and IW buffers.

(2) Influence of oxygen fugacity on $D_{\text{N}}^{\text{K-holl}/\text{Fluid}}$. In the NH_4NO_3 -bearing run, no nitrogen is detected in the K-hollandite based on the EPMA and Raman spectroscopy, in contrast to the high nitrogen content of 6553 ± 813 ppm in the K-hollandite of the NH_3 -bearing run at the same P-T condition (Fig. R3-2 c). This suggests that high oxygen fugacity in the NH_4NO_3 -bearing run, as indicated by the presence of hematite, stabilizes N_2 over reduced nitrogen species, and nitrogen is stored in K-hollandite mainly as ammonium (Fig. R3-2 c). This contrast indicates that high oxygen fugacity may reduce the nitrogen partition coefficient between K-hollandite and fluid, depending on N species in fluid.

Anyway, our study provides nitrogen partition coefficient between K-hollandite and fluid of the NH_3 -bearing runs corresponding to the oxygen fugacity between NNO and IW buffers similar to the subduction zone redox conditions³⁴.

Fig. R3-2 (a) Fe contents in garnet in different systems. (b) Pressure dependence of the substitution of $\text{NaAlSi}_2\text{O}_6$ to $\text{CaMgSi}_2\text{O}_6$ compositions, and the substitution of $\text{NaFe}^{3+}\text{Si}_2\text{O}_6$ to $\text{NaAlSi}_2\text{O}_6$ occurring in NH_4NO_3 -bearing run. (c) Raman spectroscopy of K-hollandite with different nitrogen contents. Peaks in the range of $2700\text{--}3400\text{ cm}^{-1}$ are the signal of ammonium. Asterisks indicate that the peaks are influenced by the contamination of resin.

The mass balance calculations appear to assume that exactly 8 wt% of a fluid was added to oxide powders to generate the starting composition. It is not possible to add exactly 8 wt % of fluid to a capsule and then have 100 % retention with welding, and given the centrality of this number, discussion of the precision and accuracy of this number needs to be included.

Reply: Thanks for the pointing out this. We have corrected this and discussed the precision and accuracy in the revision (**Methods lines 287-293; Supplementary Text 4; Supplementary Table 1**), and we used the corrected the fluid mass fractions of each NH_3 -bearing run in the following Table R3-1 to calculate nitrogen partition coefficients for the NH_3 -bearing runs.

For the NH_3 -bearing runs, a NH_3 (25 wt%)- H_2O solution was used as the water and nitrogen source. The added volume of NH_3 solution was calculated by the ideal mass of NH_3 solution and its density (0.9). The difference between the capsule weight before the addition of NH_3 solution and that after the gas-tightness test is considered as the mass of NH_3 solution in the capsule. By comparison with the ideal mass of NH_3 solution that should be loaded in, we can estimate the maximum uncertainty induced by the loading and welding processes. The fluid retention in the capsule ranges from ~77 to 94%. This indicates that ~6.2-7.5 wt% fluid has been added to the system. However, the concentration of NH_3 in fluid should not change greatly due to the consistent P-T dependence of the nitrogen content in K-hollandite, despite variations in fluid retention among different runs. For example, the 10 GPa-1100 °C run with a fluid

retention of 94% has the lowest nitrogen content in K-hollandite, while the 12 GPa-1000 °C run with a fluid retention of 77% has the second highest nitrogen content in K-hollandite. Therefore, the fluid mass fraction of each run (Table R3-1) was concurrently corrected for both water and NH₃ during the calculation of partition coefficients.

Table R3-1 Detailed fluid retention for each NH₃-bearing run.

NH₃ Runs	Fluid retention (%)	Corrected H₂O (wt%)	Corrected NH₃ (wt%)
10GPa-1000°C	78	4.7	1.6
10GPa-1100°C	94	5.7	1.9
10.5GPa-800°C	90	5.4	1.8
11GPa-800°C	77	4.6	1.5
11GPa-900°C	85	5.1	1.7
11GPa-1000°C	78	4.7	1.6
11GPa-1100°C	77	4.6	1.5
12GPa-1000°C	77	4.6	1.5

The manuscript states that carbonates were used as sources of Ca, Na, and K. Typically these are fired in air to decarbonate, but firing will oxidize the FeO (potentially leading to oxidation of N in experiments). If not fired, then clearly CO₂ will be a part of the system. More detail regarding starting composition preparation needs to be included. Was a LOI taken for the starting oxide composition to verify that it adds negligible water or CO₂?

Reply: Thanks for the suggestion. We have dried the carbonates to remove water and decarbonate. To avoid being oxidized, the FeO was added to the mixture after decarbonation. We did not take a LOI, but the whole preparation process is generally applied in this field and should remove CO₂ and absorbed water. We have added the detailed descriptions about the preparation of starting materials in this revision (**Methods lines 278-288**).

To remove absorbed water, SiO₂, TiO₂, Al₂O₃, and MgO powders were heated at 1000 °C for 12 h, Na₂CO₃ and K₂CO₃ powders were fired at 100 °C for 5 h, and CaCO₃ powder was held at 200 °C for 5 h. These chemical powders were mixed and ground in ethanol in an agate mortar for at least 1 h and subsequently dried at room temperature, after which the dried mixture was decarbonated at 1000 °C for 10 h. Finally, the FeO powder was mixed with the decarbonated mixture and ground in ethanol in an agate mortar for at least 1 h, after which the final mixture was dried at room temperature. All

the dried starting materials were stored in a vacuum oven at 100 °C for at least 24 h before being loaded into the Au capsules. NH₃ (25 wt%)-H₂O solution and NH₄NO₃ were used as nitrogen sources for the eight NH₃-bearing runs and one NH₄NO₃-bearing run, respectively.

I was lost in the section regarding diamonds and suggest it be removed to provide more focus on the essential data. Similarly, I had a hard time understanding the uncertainties regarding the N flux calculations. More robust number to focus on may be simply that fraction of N retained in the slab to the point of phengite breakdown is released.

Reply: Thanks for your suggestion. We have removed the section about diamond formation. We have adjusted the paragraphs, and focused more on sedimentary nitrogen subduction efficiency passing through the phengite to K-hollandite transition (**Lines 188-239**). To make the nitrogen flux calculation clearer, we have re-organized language and paragraphs in this revision (**Lines 241-257**). We also modified Fig. 4b by adding the values of sedimentary nitrogen subduction efficiency.

I commend the authors for their attempt to calculate the mass and composition of the supercritical fluid in their experiments, but insufficient details were given for the reader to evaluate this work in the manuscript. It may be helpful to publish this work as a separate manuscript to provide space and focus for this work to be understood and evaluated.

Reply: Thanks very much for your recommendation. We applied machine learning to calculate the mass and composition of the supercritical fluid in our experiments. This method part was in the original Supplementary Text 3 (**now Lines 138-147; Methods lines 371-402; Supplementary Text 5**), which was specifically reviewed by Reviewer #1.

According to the suggestion of Reviewer #1, we have added some descriptions about datasets, the processes for building the models, uncertainties of the models, and the performances of different algorithms in the revised manuscript (**Lines 138-147; lines 371-402**). We also have provided the description about the principles of the algorithms in the revised **Supplementary Text 5**.

The comparison with the Ono phengite-out determination is important, but the conclusion that N (at relatively low concentrations) can shift this boundary by ~2 GPa

is weak. A comparison of bulk system chemistries and experimental approaches need to be considered. What is the pressure calibration of the presses and assemblies used in these studies? A more firm conclusion could be derived if the same starting composition and same press+assembly was used for N-bearing and N-free experiments.

Reply: Yes, we have discussed the bulk system chemistry and experimental approaches of our study and Ono. (1998)³⁵ in this revision (**Lines 92-100; Methods lines 304-308; Supplementary Fig. 4**).

The compositions of the starting materials we used is the same as that used by Ono. (1998), and the only difference is that we added nitrogen in the system. Therefore, the impact of compositions except nitrogen can be excluded. The relationships between the press load and the sample pressure were calibrated following phase transformations (Fig. R3-3a): Bi (I-II, 2.55 Pa), Bi (III-V, 7.7 Pa), Pb (13.6 Pa), ZnS (15.6 GPa) at room temperature and **quartz-coesite** (3.5 GPa), **coesite-stishovite** (9.9 GPa), **olivine-wadsleyite** (14.5 GPa) at 1400 °C. Under our experimental conditions, the difference between the calibrations at 1400 °C and room temperature is within 0.4 GPa. The phase transitions used for pressure calibration in Ono. (1998) are as follow: **SiO₂ (quartz-coesite)** at 3.2 GPa at 1200 °C, Fe₂SiO₄ (α - γ) at 5.8 and 6.3 GPa at 1200 °C and 1400 °C, **SiO₂ (coesite-stishovite)** at 9.7 GPa at 1400°C, and **Mg₂SiO₄ (α - β)** at 14.3 GPa and 15.1 GPa at 1400 °C and 1600 °C. There are three same phase transitions in pressure range of ~3-15 GPa used for pressure calibration in our study and Ono. (1998). Therefore, uncertainty induced by pressure calibration should be small. In addition, the substitution of (Mg, Fe²⁺) + Si = Al^{IV} + Al^{VI} in phengite displays good correlation with pressure in the pelitic system (Fig. R3-3b), and the Si, Al, Mg, and Fe contents of phengite in our work and Ono's work almost display the same pressure dependence. It further demonstrates that the uncertainty of pressure induced by pressure calibration is negligible and should not influence our conclusions.

Fig. R3-3 (a) Pressure calibration using 18/11 assemblies in the 2500-ton Cubic-style multi-anvil apparatus UHP-2500 at Guangzhou Institute of Geochemistry, Chinese Academy of Sciences. The pressure-force relations of the best fit functions are $P = -2.164300 \times 10^{-6} F^2 + 1.264096 \times 10^{-2} F + 7.852733 \times 10^{-2}$ for 25 °C (black curve), and $P = -2.504000 \times 10^{-6} F^2 + 1.301033 \times 10^{-2} F - 2.898133 \times 10^{-1}$ for 1400 °C (red curve), where P is pressure (GPa) and F is force (tons). (b) Correlation between the substitution of $(\text{Mg}, \text{Fe}^{2+}) + \text{Si} = \text{Al}^{\text{IV}} + \text{Al}^{\text{VI}}$ in phengite and pressure.

Please discuss the choice of functional form used for the Eq on line 163 (and I presume that R2 should be 0.92 not 92).

Reply: Yes, it was 0.92. In this revision, we use the equation of $Y = a/T + b \cdot P/T + c$ to fit the P-T dependence of $\log(D_N^{\text{K-holl/Fluid}})$ (Lines 157-162).

The equilibrium constant (K) of nitrogen partitioning between mineral and fluid can be expressed as³⁶:

$$\ln K = \ln \left(\frac{a_N^{\text{Mineral}}}{a_N^{\text{Fluid}}} \right) = \ln \left(\frac{C_N^{\text{Mineral}} \gamma_N^{\text{Mineral}}}{C_N^{\text{Fluid}} \gamma_N^{\text{Fluid}}} \right) = -\frac{\Delta G}{RT} = -\frac{\Delta H}{RT} + \frac{\Delta S}{R} - \frac{P\Delta V}{RT}$$

Therefore, the $\log(D_N^{\text{Mineral/Fluid}})$ can be expressed as:

$$\ln D = \ln \left(\frac{C_N^{\text{Mineral}}}{C_N^{\text{Fluid}}} \right) = \ln K - \ln \left(\frac{\gamma_N^{\text{Mineral}}}{\gamma_N^{\text{Fluid}}} \right) = -\frac{\Delta H}{RT} + \frac{\Delta S}{R} - \frac{P\Delta V}{RT} - \ln \left(\frac{\gamma_N^{\text{Mineral}}}{\gamma_N^{\text{Fluid}}} \right)$$

$$\log D = \frac{\ln D}{\ln 10} = -\frac{\Delta H}{RT \ln 10} + \frac{\Delta S}{R \ln 10} - \frac{P\Delta V}{RT \ln 10} - \log \left(\frac{\gamma_N^{\text{Mineral}}}{\gamma_N^{\text{Fluid}}} \right)$$

a , and γ are thermodynamic activity and activity coefficient of N, respectively. ΔG , ΔH , ΔS , and ΔV are the changes of Gibbs free energy, enthalpy, entropy, and volume for the exchange reaction, respectively. R is the gas constant.

Multiple linear regression of the experimental data yields the following equation:

$$\log\left(D_N^{\text{K-holl/Fluid}}\right) = \frac{-2996.99(\pm 1365.49)}{T} + \frac{495.67(\pm 113.53)P}{T} - 3.44(\pm 0.51)$$

($R^2=0.84$, $p\text{-value}=0.01$) (2)

An important possibility to consider is that AOC and sediments may have water-rich fluid pass through them relating to the breakdown of serpentinites. The analysis of this paper appears to neglect this possibility.

Reply: Thanks for the comment. In this revision, we have discussed the nitrogen preservation in phengite considering fluid existing, and thereby revised the discussion about sedimentary nitrogen subduction efficiency passing through the phengite to K-hollandite transition in this revision (**Lines 228-239; Supplementary Table 4; Supplementary Fig. 3**).

Without considering fluid supplied from other sources, we estimate the maximum values of the initial sedimentary nitrogen that can pass through the phengite to K-hollandite transition for the cold IBM slab of 30% and the warm CA slab of 15%. It should be noted that any breakdown or dehydration of phengite at the depth before the phengite to K-hollandite transition¹⁰ would decrease the estimated nitrogen deep subduction efficiency; and any reaction between phengite and fluids derived from the subducting slab, such as those from the serpentinite, would also decrease the estimated nitrogen deep subduction efficiency. It depends on the amount of fluid in the slabs, especially supplied by serpentine dehydration subject to geotherms¹¹⁻¹³. We take the cold Izu-Bonin-Mariana (IBM) slab and warm Central America (CA) slab as representative slabs in our discussion (Fig. R3-4). In the cold IBM slab, serpentine-bearing peridotite transfers to the Phase A-bearing peridotite and almost no fluid released¹¹, so that 100% nitrogen can be preserved in phengite and carried to the K-hollandite filter. On the other hand, serpentinitized slab mantle completely dehydrates at ~6 GPa in the warm CA slab, which will cause a significant nitrogen loss from phengite. In this scenario, assuming a serpentinitized slab mantle with a thickness of 2 km and water content of 2 wt%¹⁴, the nitrogen preservation in phengite is calculated to be only ~12.3% based on the reported value of $D_N^{\text{Phe/Fluid}}$ at 5.5 and 6.3 GPa^{15,16}. This results in only ~7% of slab sedimentary nitrogen transported to the phengite to K-hollandite transition depth, and only ~2% of slab sedimentary nitrogen passing through the phengite to K-hollandite transition in the CA slab. If with fluid from other sources poor

in nitrogen, such as the slab oceanic crust, the nitrogen preservation in phengite to the K-hollandite filter of both cold and warm slabs can be even smaller, especially considering that $D_N^{\text{Phe/Fluid}}$ at 10 GPa is much smaller.

Fig. R3-4 Illustration of nitrogen preservation of phengite considering serpentinite dehydration. P-T path of the slab Moho is from D80 model ¹². The phase relations of water saturated peridotite are displayed by black dashed lines ¹¹. In the cold slab (Izu-Bonin-Mariana, cyan line), antigorite directly transfers to Phase A and no fluid will be released up to the depth of mantle transition zone ¹¹. In the warm slab (Central American, orange line), antigorite completely dehydrates and significant fluid will be produced, which will cause nitrogen loss from phengite. 100% and 12.3% are the estimated nitrogen preservations in phengite of the cold slab and warm slab, respectively.

Please discuss the K-alpha peak position of N in BN vs that in your samples.

Reply: The N K α peak positions of BN (147.56 ± 0.01 mm) and our sample (147.19 ± 0.29 mm) are similar within the error range (Fig. R3-5). We have provided the detailed introduction of nitrogen measurement and the accuracy in this revised

manuscript (Methods lines 327-342; Supplementary Fig. 5).

Fig. R3-5 Comparison of the N K-alpha peak between BN and K-hollandite. Baseline correction for the N K-alpha peak of (a) BN and (b) K-hollandite (NH-8). Peakfitting of the N K-alpha peak of (c) BN and (d) K-hollandite with Gaussian + Lorentzian function. The N K-alpha peak positions of BN and K-hollandite are 147.56 ± 0.01 mm and 147.19 ± 0.29 mm, respectively.

References:

1. Petrelli, M., Caricchi, L. & Perugini, D. Machine Learning Thermo-Barometry: Application to Clinopyroxene-Bearing Magmas. *J. Geophys. Res.: Solid Earth* **125**, e2020JB020130 (2020).
2. Huang, W. H. *et al.* Estimating ferric iron content in clinopyroxene using machine learning models. *Am. Mineral.* **107**, 1886–1900 (2022).
3. Lei, J., Sen, S., Li, Y. & ZhangZhou, J. Carbon in the deep upper mantle and transition zone under reduced conditions: Insights from high-pressure experiments and machine learning models. *Geochim. Cosmochim. Acta* **332**, 88–102 (2022).
4. Mitchell, E. C. *et al.* Nitrogen sources and recycling at subduction zones: Insights

- from the Izu-Bonin-Mariana arc. *Geochem. Geophys. Geosyst.* **11**, Q02X11 (2010).
5. Labidi, J. *et al.* Recycling of nitrogen and light noble gases in the Central American subduction zone: Constraints from $^{15}\text{N}^{15}\text{N}$. *Earth Planet. Sci. Lett.* **571**, 117112 (2021).
 6. Li, K. & Li, L. Nitrogen enrichments in sheeted dikes and gabbros from DSDP/ODP/IODP Hole 504B and 1256D: Insights into nitrogen recycling in Central America and global subduction zones. *Geochim. Cosmochim. Acta* **335**, 197–210 (2022).
 7. Li, K. & Li, L. Nitrogen enrichment in the altered upper oceanic crust: A new perspective on constraining the global subducting nitrogen budget and implications for subduction-zone nitrogen recycling. *Earth Planet. Sci. Lett.* **602**, 117960 (2023).
 8. Hilton, D. R., Fischer, T. P. & Marty, B. Noble Gases and Volatile Recycling at Subduction Zones. *Rev. Mineral. Geochem.* **47**, 319–370 (2002).
 9. Schmidt, M. W., Vielzeuf, D. & Auzanneau, E. Melting and dissolution of subducting crust at high pressures: the key role of white mica. *Earth Planet. Sci. Lett.* **228**, 65–84 (2004).
 10. Halama, R., Bebout, G. E., Marschall, H. R. & John, T. Fluid-induced breakdown of white mica controls nitrogen transfer during fluid–rock interaction in subduction zones. *International Geology Review* **59**, 702–720 (2017).
 11. Schmidt, M. W. & Poli, S. Devolatilization During Subduction. in *Treatise on Geochemistry* 669–701 (Elsevier, 2014).
 12. Syracuse, E. M., van Keken, P. E. & Abers, G. A. The global range of subduction

- zone thermal models. *Phys. Earth Planet. Inter.* **183**, 73–90 (2010).
13. Shirey, S. B., Wagner, L. S., Walter, M. J., Pearson, D. G. & van Keken, P. E. Slab Transport of Fluids to Deep Focus Earthquake Depths-Thermal Modeling Constraints and Evidence From Diamonds. *AGU Adv* **2**, e2020AV000304 (2021).
 14. Van Keken, P. E., Hacker, B. R., Syracuse, E. M. & Abers, G. A. Subduction factory: 4. Depth-dependent flux of H₂O from subducting slabs worldwide. *J. Geophys. Res.: Solid Earth* **116**, B01401 (2011).
 15. Kupriyanov, I. N. *et al.* Nitrogen fractionation in mica metapelite under hot subduction conditions: Implications for nitrogen ingassing to the mantle. *Chem. Geol.* **628**, 121476 (2023).
 16. Sokol, A. G. *et al.* Nitrogen storage capacity of phengitic muscovite and K-cymrite under the conditions of hot subduction and ultra high pressure metamorphism. *Geochim. Cosmochim. Acta* **355**, 89–109 (2023).
 17. Deng, X. *et al.* Compositional and thermal state of the lower mantle from joint 3D inversion with seismic tomography and mineral elasticity. *Proc. Natl. Acad. Sci. U.S.A.* **120**, e2220178120 (2023).
 18. Yu, C., Goes, S., Day, E. A. & Van Der Hilst, R. D. Seismic evidence for global basalt accumulation in the mantle transition zone. *Sci. Adv.* **9**, eadg0095 (2023).
 19. Yan, J., Ballmer, M. D. & Tackley, P. J. The evolution and distribution of recycled oceanic crust in the Earth's mantle: Insight from geodynamic models. *Earth Planet. Sci. Lett.* **537**, 116171 (2020).
 20. Stachel, T., Harris, J. W., Brey, G. P. & Joswig, W. Kankan diamonds (Guinea) II:

- lower mantle inclusion parageneses. *Contrib. Mineral. Petrol.* **140**, 16–27 (2000).
21. Plá Cid, J., Nardi, L. V. S., Plá Cid, C., Enrique Gisbert, P. & Balzaretto, N. M. Acid compositions in a veined-lower mantle, as indicated by inclusions of (K,Na)-Hollandite+SiO₂ in diamonds. *Lithos* **196–197**, 42–53 (2014).
 22. Bulanova, G. P. *et al.* Mineral inclusions in sublithospheric diamonds from Collier 4 kimberlite pipe, Juina, Brazil: subducted protoliths, carbonated melts and primary kimberlite magmatism. *Contrib. Mineral. Petrol.* **160**, 489–510 (2010).
 23. Smart, K. A., Tappe, S., Stern, R. A., Webb, S. J. & Ashwal, L. D. Early Archaean tectonics and mantle redox recorded in Witwatersrand diamonds. *Nat. Geosci.* **9**, 255–259 (2016).
 24. Hirao, N., Ohtani, E., Kondo, T., Sakai, T. & Kikegawa, T. Hollandite II phase in KAlSi₃O₈ as a potential host mineral of potassium in the Earth's lower mantle. *Phys. Earth Planet. Inter.* **166**, 97–104 (2008).
 25. Busigny, V., Cartigny, P. & Philippot, P. Nitrogen isotopes in ophiolitic metagabbros: A re-evaluation of modern nitrogen fluxes in subduction zones and implication for the early Earth atmosphere. *Geochim. Cosmochim. Acta* **75**, 7502–7521 (2011).
 26. Som, S. M., Catling, D. C., Harnmeijer, J. P., Polivka, P. M. & Buick, R. Air density 2.7 billion years ago limited to less than twice modern levels by fossil raindrop imprints. *Nature* **484**, 359–362 (2012).
 27. Som, S. M. *et al.* Earth's air pressure 2.7 billion years ago constrained to less than half of modern levels. *Nat. Geosci.* **9**, 448–451 (2016).
 28. Marty, B., Zimmermann, L., Pujol, M., Burgess, R. & Philippot, P. Nitrogen

- Isotopic Composition and Density of the Archean Atmosphere. *Science* **342**, 101–104 (2013).
29. Yoshioka, T., Wiedenbeck, M., Shcheka, S. & Keppler, H. Nitrogen solubility in the deep mantle and the origin of Earth's primordial nitrogen budget. *Earth Planet. Sci. Lett.* **488**, 134–143 (2018).
30. Johnson, B. & Goldblatt, C. The nitrogen budget of Earth. *Earth-Sci. Rev.* **148**, 150–173 (2015).
31. Bergin, E. A., Blake, G. A., Ciesla, F., Hirschmann, M. M. & Li, J. Tracing the ingredients for a habitable earth from interstellar space through planet formation. *Proc. Natl. Acad. Sci. U.S.A.* **112**, 8965–8970 (2015).
32. Marty, B. The origins and concentrations of water, carbon, nitrogen and noble gases on Earth. *Earth Planet. Sci. Lett.* **313–314**, 56–66 (2012).
33. Dobrzhinetskaya, L. F. & Green, H. W. Experimental studies of mineralogical assemblages of metasedimentary rocks at Earth's mantle transition zone conditions. *J. Metamorph. Geol.* **25**, 83–96 (2007).
34. Foley, S. F. A Reappraisal of Redox Melting in the Earth's Mantle as a Function of Tectonic Setting and Time. *J. Petrol.* **52**, 1363–1391 (2011).
35. Ono, S. Stability limits of hydrous minerals in sediment and mid-ocean ridge basalt compositions: Implications for water transport in subduction zones. *Journal of Geophysical Research: Solid Earth* **103**, 18253–18267 (1998).
36. Sun, C. Partitioning and Partition Coefficients. in *Encyclopedia of Geochemistry: A Comprehensive Reference Source on the Chemistry of the Earth* (ed. White, W.

M.) 1–11 (Springer International Publishing, Cham, 2017).

REVIEWER COMMENTS

Reviewer #1 (Remarks to the Author):

As stated in my review of the previous version of this manuscript, I have been asked to review the machine learning component of the manuscript. My review will focus purely on the machine learning methods, their application, and reviewing the accurate representation of the results of fluid and melt predictive models. I must confess that domain of application and the high pressure-high temp experiments are outside my areas of expertise, so I will be very focused on the machine learning aspect of this work.

As per suggestions on the previous version of the manuscript, the authors have included more details on the machine learning method in the main manuscript. The authors have also addressed all of my previous suggestions to improve the accuracy and clarity of the machine learning component of this manuscript.

One minor point: Line 392-394: It is unclear to me what this sentence is stating. I assume the authors mean that since their dataset is small, the final fluid and melt models made available to researchers will be trained on all the data. If this is the correction interpretation, then I recommend modifying the sentence to make it a bit clearer to the reader.

Overall, the machine learning component of the manuscript has improved and the authors have taken reviewer suggestions to provide key details about the machine learning method, the results, and the reproducibility of the model. I still do not feel entirely comfortable providing an accept or reject because the domain of application and the high pressure-high temp experiments are outside my areas of expertise. I will however state that the machine learning component of this paper is well described and the authors have provided sufficient information on the choice of algorithm, the comparison of accuracy of results between models, and the reproducibility of the machine learning algorithm.

Reviewer #1 (Remarks on code availability):

I was able to download and run the code in my local machine. The authors provide enough instruction to install and run the code.

Reviewer #2 (Remarks to the Author):

I read the manuscript again and I am happy with all the improvements and the detailed responses to my concerns. I recommend publication now!

Reviewer #3 (Remarks to the Author):

Huang et al present a revised draft of their manuscript focused on the partitioning of N in sediment-fluid systems containing phengite and K-hollandite. I commend the authors on completing new experiments, additional analysis regarding their peak centers, and discussion regarding oxygen fugacity.

I still have issues with the following:

1) The additional experiment conducted with a more oxidized starting fluid (NH_4NO_3) now demonstrates that oxygen fugacity likely plays a role in N partitioning at ~ 10 GPa in slab system, and yet the effect of redox remains unaccounted for in the parameterization of the data or application of the parameterization to modeling the fate of N in slab systems. Constraining oxygen fugacity to a 4 orders-of-magnitude range (IW to NNO) is not sufficient when developing predictions to natural systems. In short, the treatment of redox remains insufficient.

2) Nitrogen is shown to be incompatible to similar degrees in both phengite and K-hollandite (both minerals incorporate ~ 1000 ppm N in experiments), but it is argued that the preferential incorporation of N into phengite stabilizes it over K-hollandite by 2 GPa. Please further discuss this observation and seek support for other elements providing a similar degree of stabilization at these low concentrations. Perhaps the impact of N on the phengite/K-Hollandite boundary relates to changes in fluid chemistry? Although, the simple effect of adding N to a fluid should be to dilute H_2O and destabilize hydrous minerals over anhydrous ones.

3) Wavescans are now reports for the standard and an experimental mineral (K-hollandite). The peak center is provided for the K-hollandite, along with an uncertainty, but I cannot find any description of how the uncertainty was derived. The peak center depends on the background model, and it is stated that an exponential model is used, but the red line Fig S5 B seems to have a different functional form than described in the text. Backgrounds are extremely important here as signals are highly impacted by the background, as down in Fig S5.

A response to Reviewer

Reply to the comments from Reviewer #1:

Reviewer #1 (Remarks to the Author):

As stated in my review of the previous version of this manuscript, I have been asked to review the machine learning component of the manuscript. My review will focus purely on the machine learning methods, their application, and reviewing the accurate representation of the results of fluid and melt predictive models. I must confess that domain of application and the high pressure-high temp experiments are outside my areas of expertise, so I will be very focused on the machine learning aspect of this work.

As per suggestions on the previous version of the manuscript, the authors have included more details on the machine learning method in the main manuscript. The authors have also addressed all of my previous suggestions to improve the accuracy and clarity of the machine learning component of this manuscript.

Reply: Thanks for the positive comments.

One minor point: Line 392-394: It is unclear to me what this sentence is stating. I assume the authors mean that since their dataset is small, the final fluid and melt models made available to researchers will be trained on all the data. If this is the correction interpretation, then I recommend modifying the sentence to make it a bit clearer to the reader.

Reply: Yes, this is the correction interpretation. We have modified (Lines 424-425).

Overall, the machine learning component of the manuscript has improved and the authors have taken reviewer suggestions to provide key details about the machine learning method, the results, and the reproducibility of the model. I still do not feel entirely comfortable providing an accept or reject because the domain of application and the high pressure-high temp experiments are outside my areas of expertise. I will however state that the machine learning component of this paper is well described and the authors have provided sufficient information on the choice of algorithm, the comparison of accuracy of results between models, and the reproducibility of the machine learning algorithm.

Reply: Thanks for the positive comments.

Reviewer #1 (Remarks on code availability):

I was able to download and run the code in my local machine. The authors provide enough instruction to install and run the code.

Reply: Thanks for the positive comments.

Reply to the comments from Reviewer #2:

Reviewer #2 (Remarks to the Author):

I read the manuscript again and I am happy with all the improvements and the detailed responses to my concerns. I recommend publication now!

Reply: Thanks for the helpful comments and support for the publication of our paper.

Reply to the comments from Reviewer #3:

Reviewer #3 (Remarks to the Author):

Huang et al present a revised draft of their manuscript focused on the partitioning of N in sediment-fluid systems containing phengite and K-hollandite. I commend the authors on completing new experiments, additional analysis regarding their peak centers, and discussion regarding oxygen fugacity.

Reply: We sincerely thank you very much for your encouragements and helpful comments listed below. As you can see, we have now fully considered your concerns, and your concerns did help us improve the manuscript a lot!

I still have issues with the following:

1) The additional experiment conducted with a more oxidized starting fluid (NH_4NO_3) now demonstrates that oxygen fugacity likely plays a role in N partitioning at ~ 10 GPa in slab system, and yet the effect of redox remains unaccounted for in the parameterization of the data or application of the parameterization to modeling the fate of N in slab systems. Constraining oxygen fugacity to a 4 orders-of-magnitude range (IW to NNO) is not sufficient when developing predictions to natural systems. In short, the treatment of redox remains insufficient.

Reply: We sincerely appreciate these comments, which has helped us discuss the redox conditions of our samples more deeply.

As you know, the control of oxygen fugacity during the multi-anvil experiments and the estimation of oxygen fugacity after the experiments are rather challenging. We therefore did not control the sample oxygen fugacity using any external oxygen fugacity buffers, but we relied on the addition of NH₃-H₂O solution to create a relatively reducing condition.

To narrow down our previous estimates of oxygen fugacity in the range of IW to NNO, we have compared our experiments with previous experiments conducted on pelite at 3-8 GPa by Sokol et al. (2023), who also added reduced fluids (C₃H₆N₆, which decomposes into NH₃, N₂ and other volatiles) into Au capsules as the source of nitrogen. By analyzing the NH₃/N₂ ratios in their fluids after the experiments, Sokol et al. (2023) estimated that the oxygen fugacity of their experiments starting with C₃H₆N₆ was about 3-4 log units below the NNO buffer.

As we noticed in our last manuscript, the FeO content of Cpx is related to the oxygen fugacity. Based on all the experiments with reported oxygen fugacity and Cpx composition from Sokol et al. (2023), we found a positive correlation between the FeO content of Cpx and oxygen fugacity. Using the FeO contents of our Cpx, we estimated that the oxygen fugacity of our NH₃-bearing samples should be 2.7-3.1 log units below the NNO buffer (see Fig. R1 attached below).

Considering that we added NH₃-H₂O in our starting materials as the source of nitrogen, we concluded that the oxygen fugacity of our NH₃-bearing runs should be ~3 log units below the NNO buffer but above the IW buffer (due to the lack of metallic iron).

The oxygen fugacity (NNO to NNO-2) of the subducted slabs (e.g., Foley 2011, J. Petrol.) should be higher than the estimated oxygen fugacity for our experiments. Therefore, the actual $D_N^{K-holl/Fluid}$ that could be applied to subduction zones should be lower than the values obtained from our NH₃-bearing experiments, which would then more support our conclusions that an insignificant proportion of sedimentary N in the slab has been subducted into the lower mantle.

We have now added the above information in our revised main text (**Lines 197-**

218; 261-263; 397-401) and **Supplementary Fig. S4**. We hope that Reviewer#3 is now satisfied with our present estimates of sample oxygen fugacity.

Fig R1. Correlation between the FeO (wt%) in clinopyroxene and the oxygen fugacity. Data from Sokol et al. (2023).

2) Nitrogen is shown to be incompatible to similar degrees in both phengite and K-hollandite (both minerals incorporate ~1000 ppm N in experiments), but it is argued that the preferential incorporation of N into phengite stabilizes it over K-hollandite by 2 GPa. Please further discuss this observation and seek support for other elements providing a similar degree of stabilization at these low concentrations. Perhaps the impact of N on the phengite/K-Hollandite boundary relates to changes in fluid chemistry? Although, the simple effect of adding N to a fluid should be to dilute H₂O and destabilize hydrous minerals over anhydrous ones.

Reply: Thanks for these very useful comments. In the revised manuscript, we have

now given more explanation about the extension of phengite stability field.

Our previous studies showed that ammonium (NH_4^+) in the interlayer of phengite can form hydrogen bonding with the basal oxygen of the tetrahedra (Yang et al. 2017). Due to such hydrogen bonding, the presence of ~2000 ppm ammonium can hinder the softening of phengite's lattice at high temperatures and thus favor the stabilization of phengite (Huang et al. 2021). Based on the available data in previous studies on muscovite, Huang et al. (2023) further concluded that even trace amounts of NH_4^+ can enhance the thermal stability of muscovite, e.g., less than 1000 ppm (Pöter et al. 2004, 2007). Therefore, the extended stability field of phengite in the NH_3 -bearing system should be due to the presence of NH_4^+ in phengite.

Following your suggestions, we have fortunately found other evidence to show that elements with low concentrations can affect the phase transitions. For example, the presence of ~1500-4000 ppm fluorine in olivine and wadsleyite can increase the pressure for the olivine to wadsleyite transition by ~2.6 GPa compared to volatile-free systems (Grützner et al. 2018).

We have now added the above information in the revised manuscript (**Lines 103-115**). In addition, nitrogen in fluid would dilute H_2O , which should destabilize the hydrous phengite, inconsistent with our findings. So, we do not believe the pressure extension of phengite to K-hollandite transition is due to fluid chemistry modified with nitrogen.

3) Wavescans are now reports for the standard and an experimental mineral (K-hollandite). The peak center is provided for the K-hollandite, along with an uncertainty, but I cannot find any description of how the uncertainty was derived. The peak center depends on the background model, and it is stated that an exponential model is used, but the red line Fig S5 B seems to have a different functional form than described in the text. Backgrounds are extremely important here as signals are highly impacted by the background, as shown in Fig S5.

Reply: We thank you very much for pointing out the problem of Fig. S5, and we also express our regrets that in our last manuscript, we showed an old scan of N $K\alpha$

peak of one K-hollandite sample.

We did use an exponential model to subtract the baseline. Now, in the revised manuscript, we have showed the updated scan of N $K\alpha$ peak of our one K-hollandite sample (Lines 361-363; Supplementary Text 5; Supplementary Fig. 3).

The N $K\alpha$ peak of the BN standard was directly scanned. To determine the position of the N $K\alpha$ peak of the K-hollandite samples, we measured the intensities at the discrete wavenumbers using long counting times. The backgrounds were corrected using an exponential model (see Fig. R2 below), and the peak positions of the N $K\alpha$ peaks were obtained through Gaussian + Lorentzian peak-fitting. We repeated the background-correction and peak-fitting for multiple times, and each time different data points were selected to fit the exponential baseline. This procedure allowed us to obtain multiple corrected N $K\alpha$ peaks and their corresponding peak positions. The average value and its standard deviation of the peak positions were considered as the final peak position and its uncertainty, respectively. The N $K\alpha$ peak of BN was located at 147.54 ± 0.003 mm, while the N $K\alpha$ peak of K-hollandite was at 147.21 ± 0.01 mm.

Fig R2. Comparison of N $K\alpha$ peaks between BN standard and K-hollandite. Baseline correction using an exponential model for the N $K\alpha$ peak of (a) BN and (b) K-hollandite (NH-8). Peak fitting of the corrected N $K\alpha$ peak of (c) BN and (d) K-hollandite with a Gaussian + Lorentzian function. The N $K\alpha$ peak positions of BN and K-hollandite are 147.54 ± 0.003 mm and 147.21 ± 0.01 mm, respectively.

References

- Foley, S.F. (2011) A Reappraisal of Redox Melting in the Earth's Mantle as a Function of Tectonic Setting and Time. *Journal of Petrology*, 52, 1363–1391.
- Grützner, T., Klemme, S., Rohrbach, A., Gervasoni, F., and Berndt, J. (2018) The effect of fluorine on the stability of wadsleyite: Implications for the nature and depths of the transition zone in the Earth's mantle. *Earth and Planetary Science Letters*, 482, 236–244.
- Huang, W.H., Yang, Y., Qi, Z.M., Liu, W.D., Wang, Z.P., Liu, Y., and Xia, Q.K. (2021) Ammonium Impacts on Vibrations of Hydroxyl and Lattice of Phengite at High Temperature and High Pressure. *Journal of Earth Science*, 32, 1278–1286.
- Huang, W.H., Yang, Y., Gui, W.B., Liu, J., Lv, Y.F., Wang, Z.P., and Xia, Q.K. (2023) Nitrogen impacts on structural stability of feldspar: Constraints from high temperature and high pressure spectroscopy and machine learning. *Physics of the Earth and Planetary Interiors*, 336, 106997.
- Pöter, B., Gottschalk, M., and Heinrich, W. (2004) Experimental determination of the ammonium partitioning among muscovite, K-feldspar, and aqueous chloride solutions. *Lithos*, 74, 67–90.
- (2007) Crystal-chemistry of synthetic K-feldspar–buddingtonite and muscovite–tobelite solid solutions. *American Mineralogist*, 92, 151–165.
- Sokol, A.G., Kupriyanov, I.N., Kotsuba, D.A., Korsakov, A.V., Sokol, E.V., and Kruk, A.N. (2023) Nitrogen storage capacity of phengitic muscovite and K-cymrite under the conditions of hot subduction and ultra high pressure metamorphism.

Geochimica et Cosmochimica Acta, 355, 89–109.

Yang, Y., Busigny, V., Wang, Z., and Xia, Q. (2017) The fate of ammonium in phengite at high temperature. *American Mineralogist*, 102, 2244–2253.

REVIEWERS' COMMENTS

Reviewer #3 (Remarks to the Author):

I have reviewed the manuscripts and am happy with the revisions. I only request that the reader be alerted that the N partitioning parameterizations presented apply to reducing conditions (~NNO-3).